# Enhancer remodeling promotes tumor-initiating activity in NRF2-activated non-small cell lung cancers

Keito Okazaki[1], Hayato Anzawa [2], Zun Liu[1], Nao Ota[1], Hiroshi Kitamura[1], Yoshiaki Onodera[3], Md. Morshedul Alam [1], Daisuke Matsumaru [1], Takuma Suzuki[1], Fumiki Katsuoka[4], Shu Tadaka[4], Ikuko Motoike[4], Mika Watanabe[5], Kazuki Hayasaka [1,6], Akira Sakurada[6], Yoshinori Okada[6], Masayuki Yamamoto [4,7], Takashi Suzuki[8], Kengo Kinoshita [2,4], Hiroki Sekine [1✉] & Hozumi Motohashi [1✉]

Transcriptional dysregulation, which can be caused by genetic and epigenetic alterations, is a fundamental feature of many cancers. A key cytoprotective transcriptional activator, NRF2, is often aberrantly activated in non-small cell lung cancers (NSCLCs) and supports both aggressive tumorigenesis and therapeutic resistance. Herein, we find that persistently activated NRF2 in NSCLCs generates enhancers at gene loci that are not normally regulated by transiently activated NRF2 under physiological conditions. Elevated accumulation of CEBPB in NRF2-activated NSCLCs is found to be one of the prerequisites for establishment of the unique NRF2-dependent enhancers, among which the *NOTCH3* enhancer is shown to be critical for promotion of tumor-initiating activity. Enhancer remodeling mediated by NRF2-CEBPB cooperativity promotes tumor-initiating activity and drives malignancy of NRF2-activated NSCLCs via establishment of the NRF2-NOTCH3 regulatory axis.

[1] Department of Gene Expression Regulation, Institute of Development, Aging and Cancer, Tohoku University, Sendai 980-8575, Japan. [2] Department of System Bioinformatics, Graduate School of Information Sciences, Tohoku University, 980-8579 Sendai, Japan. [3] Department of Anatomic Pathology, Tohoku University Graduate School of Medicine, Sendai 980-8575, Japan. [4] Department of Integrative Genomics, Tohoku Medical Megabank Organization, Tohoku University, Sendai 980-8573, Japan. [5] Department of Pathology, Tohoku University Hospital, Sendai 980-8575, Japan. [6] Department of Thoracic Surgery, Institute of Development, Aging and Cancer, Tohoku University, Sendai 980-8575, Japan. [7] Department of Medical Biochemistry, Tohoku University Graduate School of Medicine, Sendai 980-8575, Japan. [8] Department of Pathology and Histotechnology, Tohoku University Graduate School of Medicine, Sendai 980-8575, Japan. ✉email: sekine@med.tohoku.ac.jp; hozumim@med.tohoku.ac.jp

Transcriptional dysregulation is an important feature of many cancers[1]. Epigenetic alterations leading to aberrant transcription have been increasingly recognized in carcinogenesis ever since the emergence of the concept of the super-enhancer. These unique enhancer regions play a central role in the maintenance of cancer cell identity and drive oncogenic transcriptional programs to which cancer cells are highly addicted[2]. Enhancer reprogramming, which is caused by the redistribution of transcription factors and the subsequent changes in transcription factor networks, drives cancer cell phenotypic drift during cancer initiation and progression[3,4]. Thus, the elucidation of a cancer cell-specific enhancer landscape and transcription factors associated with the epigenetic environment is expected to provide powerful insights into the biological nature of cancer cells.

NRF2 (Nuclear Factor Erythroid 2 Like 2; NFE2L2) is a potent transcriptional activator that coordinately regulates many cytoprotective genes and plays a central role in defense mechanisms against oxidative and electrophilic insults[5]. Upon exposure to oxidative stress or electrophiles, NRF2 escapes KEAP1-mediated degradation and activates transcription through antioxidant response elements (AREs). While increased NRF2 activity is principally beneficial for our health, a variety of incurable cancers exploit NRF2 to achieve aggressive proliferation, tumorigenesis, and therapeutic resistance. Several causes have been described for the aberrant accumulation of NRF2 in cancer cells, including somatic mutations in KEAP1 or NRF2 genes, sequestration of KEAP1 by p62/SQSTM1 and electrophilic attack of KEAP1 thiols by fumarate[6–10].

Increased NRF2 accumulation in cancer tissues is strongly correlated with poor clinical outcomes in various cancer types[7,8,11,12]. This is because persistent activation of NRF2 in cancer cells confers multiple advantages, such as increased survival due to enhanced antioxidant and detoxification capacities[13,14], increased proliferation as a result of metabolic reprogramming[15–17], protection of translational machinery from oxidative damage[18], and aggressive tumorigenesis resulting from the modulation of secretory phenotypes[19]. In particular, NRF2 mediates drug resistance by increasing the expression of many detoxification enzymes and drug transporters[20,21], resulting in the inactivation and extrusion of small-molecule anti-cancer drugs. Due to these advantages, cancer cells with persistent NRF2 activation exhibit a heavy dependence on, or addiction to, NRF2[22].

Therapeutic resistance is a major obstacle for the development of effective cancer treatments. Resistance may arise through genetic and/or epigenetic changes that are induced in cancer cells during treatment[23]. In particular, chemo- and radio-resistant tumor-initiating cells (TICs), or cancer stem cells, impede treatment efficacy, thus leading to tumor relapse[24]. Tumor-initiating abilities of cancer cells are experimentally evaluated based on their capacity to generate grossly recognizable tumors. Thus, the self-renewal capacity of TICs is not easily separated from their proliferative and survival abilities, which are strongly enhanced by NRF2, and chemo-resistant populations expressing high levels of NRF2 are often regarded as TICs[25,26]. More precisely, it remains to be elucidated whether NRF2 does more than merely enhance proliferation and survival in order to support the tumor-initiating activity of cancer cells.

In this work, we aim at clarifying whether and how NRF2 contributes to the tumor-initiating activity and the consequent malignancy of non-small cell lung cancer (NSCLC) exhibiting NRF2 addiction, recognizing that ~15% of NSCLC cases carry somatic alterations of KEAP1 gene, which are major causes of NRF2 addiction[27–29]. We conduct an unbiased approach by investigating NRF2-dependent transcriptome in NSCLC cell lines with KEAP1 mutations (NRF2-activated NSCLCs) and in those with an intact KEAP1-NRF2 system (NRF2-normal NSCLCs). We identify a battery of genes that are regulated by NRF2 specifically in NRF2-activated NSCLCs and found that these genes are accompanied by unique NRF2-dependent enhancers. CEBPB accumulation in NRF2-activated NSCLCs is found to be one of the prerequisites for the establishment of the unique enhancers, in which NOTCH3 enhancer is critical for the promotion of tumor-initiating activity. Clinical data indeed show that NOTCH3 contributes to cancer malignancy selectively in NRF2-activated NSCLCs, strongly suggesting pathological significance of the NRF2-NOTCH3 axis. The NOTCH3 enhancer generated by NRF2 in cooperation with CEBPB establishes the NRF2-NOTCH3 axis and drives malignancy of NRF2-activated NSCLCs by promoting tumor-initiating activity.

## Results

**NRF2 promotes a stem-like phenotype of NRF2-activated NSCLCs.** To clarify whether NRF2 has any active role in promoting tumor-initiating activity, which is one of the important properties for aggressive tumorigenesis (Supplementary Fig. 1a, b), we cultured three NRF2-activated NSCLC cell lines with KEAP1 mutations, A549, H460 and H2023[30], under low attachment conditions in defined stem cell medium to allow them to grow in the form of oncospheres[31]. TICs expressing stem cell markers were enriched in oncospheres growing under this condition (Supplementary Fig. 1c). NRF2 knockdown impaired oncosphere growth (Fig. 1a–c), suggesting that when activated, NRF2 promotes a stem-like phenotype in NSCLCs.

**NOTCH3 is a unique downstream effector of NRF2 in NRF2-activated NSCLCs.** Downstream effectors of NRF2 that promote tumor-initiating activity are potential therapeutic targets for suppressing tumorigenesis and cancer recurrence. We decided to identify such factors among NRF2 downstream effectors specific to NRF2-activated NSCLCs, so that their inhibition does not interfere with the cytoprotective functions of NRF2, which play beneficial roles in cancer-bearing hosts.

We first assessed NRF2 abundance in five NSCLC cell lines: three NRF2-activated (A549, H460 and H2023) and two NRF2-normal (ABC1 and HCC4006) NSCLC cell lines (Fig. 1d). The NRF2-activated NSCLCs expressed high levels of NRF2, which could be diminished via transient transfection with an siRNA against NRF2. The two NRF2-normal NSCLCs with an intact KEAP1-NRF2 system expressed low levels of NRF2, which could be increased by treatment with an electrophile, diethylmaleate (DEM), which inhibits KEAP1 activity. To identify unique downstream effectors of NRF2 in NRF2-activated NSCLCs, we compared NRF2-dependent transcriptomes across the three NRF2-activated NSCLC cell lines and the two NRF2-normal NSCLC cell lines. In the NRF2-activated cell lines, NRF2 was knocked down using two alternative siRNAs (#28 and #30) in order to evaluate genes dependent on persistently activated NRF2 (Fig. 1e, left panel). In the NRF2-normal cell lines, NRF2 was induced using DEM to evaluate genes dependent on transiently activated NRF2 (Fig. 1e, right panel). 123 genes were downregulated by NRF2 knockdown in all three NRF2-activated cell lines (Fig. 1f), which were defined as common NRF2 downstream genes. 2271 genes were upregulated by DEM treatment in either of the NRF2-normal cell lines (Fig. 1f), which were defined as DEM-inducible genes. 87 genes that were included in both the common NRF2 downstream gene set and the DEM-inducible gene set were regarded as canonical NRF2 downstream effectors (Supplementary Fig. 2a). 36 genes included in the common NRF2 downstream gene set but not in the DEM-inducible gene set were

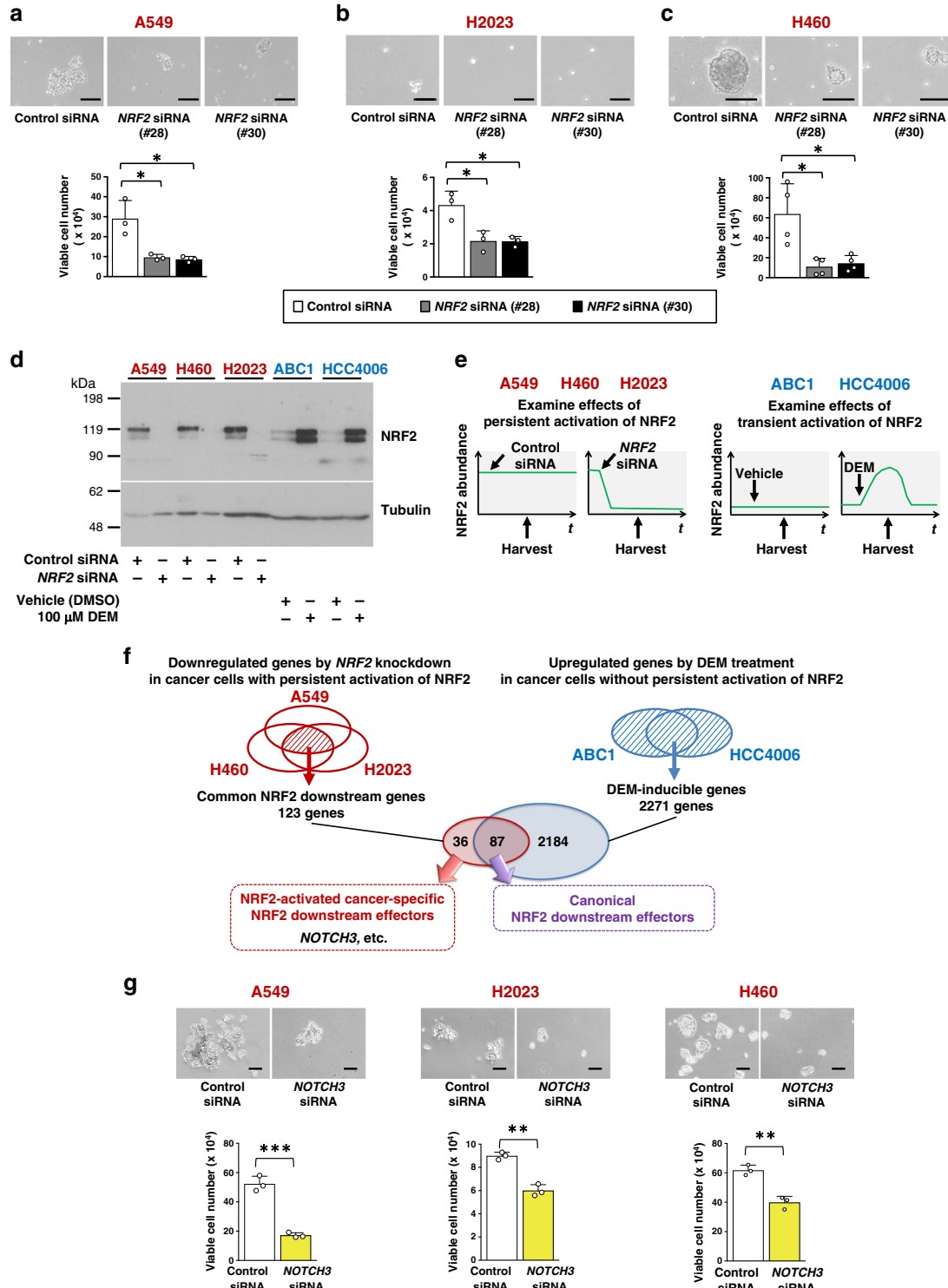

**Fig. 1 NRF2 enhances tumor-initiating activity in NRF2-activated NSCLC cell lines. a–c** Effects of *NRF2* knockdown on the oncosphere formation of A549 (**a**), H2023 (**b**), and H460 (**c**) cells. Scale bars indicate 50, 20, and 100 μm, respectively (top panels). Viable cells were counted after trypsinization (bottom panels). Average cell numbers and SD from 3 independent experiments are shown except for the experiment of H460 cells in **c**, which was independently conducted for four times. Comparison was made between two cell groups; control siRNA and each test siRNA. Two-sided Student's *t* test was performed. *$p < 0.05$. **d** Immunoblot analysis of NRF2 protein levels in NSCLC cell lines. Tubulin was used as a loading control. One representative result of three independent experiments is shown. **e** Experimental design for the preparation of RNA samples from NRF2-activated (left panels) and NRF2-normal (right panels) NSCLC cells. **f** RNA-seq analysis of NSCLC cell lines. NRF2-activated (A549, H460, and H2023) and NRF2-normal (ABC1 and HCC4006) NSCLC cells were examined. In the following Figures, NRF2-activated and NRF2-normal NSCLC cells are indicated in red and blue, respectively. **g** Effects of *NOTCH3* knockdown on the oncosphere formation of NRF2-activated NSCLC cells. Scale bars indicate 100 μm (top panels). Viable cells were counted after trypsinization (bottom panels). Average cell numbers and SD from three independent experiments are shown. Two-sided Student's *t* test was performed. **$p < 0.005$, ***$p < 0.0005$. DMSO: dimethyl sulfoxide, DEM: diethylmaleate.

regarded as NRF2-activated NSCLC-specific NRF2 downstream effectors (Supplementary Fig. 2a).

The 36 genes obtained from the cell line analysis were narrowed down to 13 based on their correlations with NRF2 activity in transcriptomic data from lung adenocarcinoma (LUAD) patients in the TCGA database (Supplementary Fig. 2b). The 13 candidates were examined for their impacts on the spheroid growth, which occurs in a cell culture mode under low attachment conditions in normal media and reflects a simple cell proliferation ability in an anchorage-independent manner (Supplementary Fig. 2c). To exclude factors that promote tumorigenesis by increasing cell proliferation ability rather than tumor-initiating ability, we selected genes whose knockdown did not suppress the spheroid growth. NOTCH3 was commonly included in the candidates obtained from the three different NRF2-activated NSCLC cell lines. NOTCH3 knockdown impaired oncosphere growth (Fig. 1g), which was consistent with previous studies describing that NOTCH3 contributes to a stem-like phenotype of NSCLC[32].

**Co-expression of NRF2 and NOTCH3 is associated with a poor prognosis in lung adenocarcinoma cases.** To examine whether the NRF2-NOTCH3 regulatory axis was also observed in clinical cases, we analyzed transcriptomic data from LUAD and squamous cell carcinoma (LUSC) patients in the TCGA database. In both LUAD and LUSC, NOTCH3 expression was elevated in the cases with high NRF2 activity (Fig. 2a). KEAP1 mutation, a major cause of increased NRF2 activity in LUAD, was uniquely enriched in LUAD cases showing high NRF2 activity with elevated NOTCH3 expression compared with other mutations (Supplementary Fig. 3a).

We next conducted a histological analysis of 41 LUAD samples from patients who had undergone surgical resection to examine a correlation of NRF2 and NOTCH3 statuses. The resected tumor tissues were stained with antibodies against NRF2 and NOTCH3 (Fig. 2b). Most of the NRF2-positive cases, which are regarded as NRF2-activated cancers, were NOTCH3-positive (Fig. 2c), which was again consistent with our observation that NOTCH3 is a downstream effector of NRF2 in NRF2-activated NSCLC cell lines. In contrast, NOTCH3-positive cases were not necessarily NRF2-positive (Fig. 2c), which suggested the presence of alternative regulators for NOTCH3. Notably, double-positive patients showed significantly poorer prognoses than the remaining cases (Fig. 2d, e and Supplementary Fig. 3b), suggesting that the combination of NRF2 and NOTCH3 makes cancers malignant and that NOTCH3 contributes to cancer malignancy when it is co-expressed with NRF2 target genes in NRF2-activated NSCLCs. Although NOTCH3 has been reported to be associated with a poor prognosis of NSCLC[33], it appears to be the NRF2-NOTCH3 axis rather than NOTCH3 itself that contributes to the malignancy of NSCLCs.

**NRF2 is a direct activator of the NOTCH3 enhancer.** To decipher a mechanism connecting persistent activation of NRF2 and NOTCH3 expression, we examined whether NRF2 directly regulates the NOTCH3 gene. We referred to a publicly-available ChIP-seq data set of A549 cells in the ENCODE database. A clear NRF2 binding peak accompanied by its heterodimeric partner molecule MAFK was observed in the intergenic region between NOTCH3 and EPHX3 genes, ~10 kbp upstream of the NOTCH3 transcription start site and 15-kb downstream of the EPHX3 transcription termination site (Fig. 3a). Three partially over-lapping ARE sequences were present in the peak area.

NRF2 binding at this site was clearly observed in all three NRF2-activated NSCLC cell lines (Fig. 3b). Because NRF2 is

known to recruit histone acetyltransferases CBP/p300 together with the SWI/SNF complex and Mediator complex[34–36], NRF2-dependent deposition of acetylated histone H3K27 (H3K27ac), an enhancer mark, was also examined at this site. The H3K27ac deposition was decreased by NRF2 knockdown in NRF2-activated NSCLC cell lines (Fig. 3c). When wild-type KEAP1 was added back to A549 cells for canceling NRF2 accumulation, the H3K27ac deposition was also decreased (Supplementary Fig. 4 and Fig. 3d). These results suggest that this intergenic NRF2-dependent enhancer is a common feature of NRF2-activated NSCLCs.

To verify that this NRF2-dependent enhancer regulates NOTCH3 expression, we disrupted the enhancer region in H460 cells by CRISPR-Cas9 genome editing technology (Supplementary Fig. 5a). To exclude off-target effects of guide RNA (gRNA), two distinct gRNAs were used to obtain mutant clones (ΔN3U H460 cells). NRF2 binding and H3K27ac deposition were abrogated in the mutated region in ΔN3U H460 cells (Fig. 3e, f), verifying successful inactivation of the enhancer. NOTCH3 expression was dramatically decreased (Fig. 3g) but did not alter the expression of EPHX3 or BRD4 (Supplementary Fig. 5b) in ΔN3U H460 cells, indicating that the intergenic NRF2 binding region serves as a major functional NOTCH3 enhancer. No compensatory upregulation of NOTCH1 or NOTCH2 was observed (Fig. 3g). In the mutant clones, NOTCH3 protein was hardly detected whereas NRF2 and its representative downstream effector genes were not affected (Supplementary Fig. 5c–e). The functional importance of this enhancer for NOTCH3 expression was also verified in A549 and H2023 cells (Supplementary Fig. 6). Thus, NOTCH3 is one of the NRF2 target genes, being directly regulated by NRF2, in NRF2-activated NSCLCs.

**The NOTCH3 enhancer is uniquely generated in NRF2-activated NSCLCs.** Because RNA-seq analysis suggested that NOTCH3 is an NRF2 target gene specifically in NRF2-activated NSCLCs, we further characterized NOTCH3 expression and NOTCH3 enhancer formation in relation to NRF2 activity in various cellular contexts.

NRF2 knockdown decreased NOTCH3 as well as NQO1, a canonical NRF2 target gene, in NRF2-activated NSCLC cells (Fig. 4a). Similar results were obtained when wild-type KEAP1 was reconstituted in NRF2-activated NSCLC cells (Supplementary Fig. 4 and Fig. 4b). In contrast, physiological transient activation of NRF2 induced by DEM treatment elevated the canonical NRF2 targets NQO1 and GCLM but not NOTCH3 in NRF2-normal NSCLC cells (Fig. 4c). Even persistent activation of NRF2 by KEAP1 knockdown in NRF2-normal NSCLC cells had no effects on NOTCH3 expression (Supplementary Fig. 7a). These results indicated that NRF2 does not regulate NOTCH3 in NRF2-normal NSCLCs. Although a reciprocal relationship between NRF2 and the NOTCH pathway was previously reported[37], NOTCH3 knockdown did not affect the expression levels of NRF2 or NQO1 in NRF2-activated NSCLC cells (Fig. 4a).

In mouse tissues, representatives of normal cells, pharmacological (CDDO-Im treatment) or genetic (Keap1 knockdown) activation of NRF2 did not induce Notch3, whereas Nqo1 was induced by both (Supplementary Fig. 7b, c). These results suggest that NRF2 does not regulate NOTCH3 in normal tissues. Thus, NOTCH3 was verified as a unique NRF2 target gene specifically in NRF2-activated NSCLCs.

In good agreement with this selective response of NOTCH3 to NRF2 activation, the NOTCH3 enhancer formation indicated by H3K27ac deposition was clearly detected in NRF2-activated NSCLCs but not in NRF2-normal NSCLCs (Fig. 4d). Transient induction of NRF2 by DEM in NRF2-normal NSCLC cells,

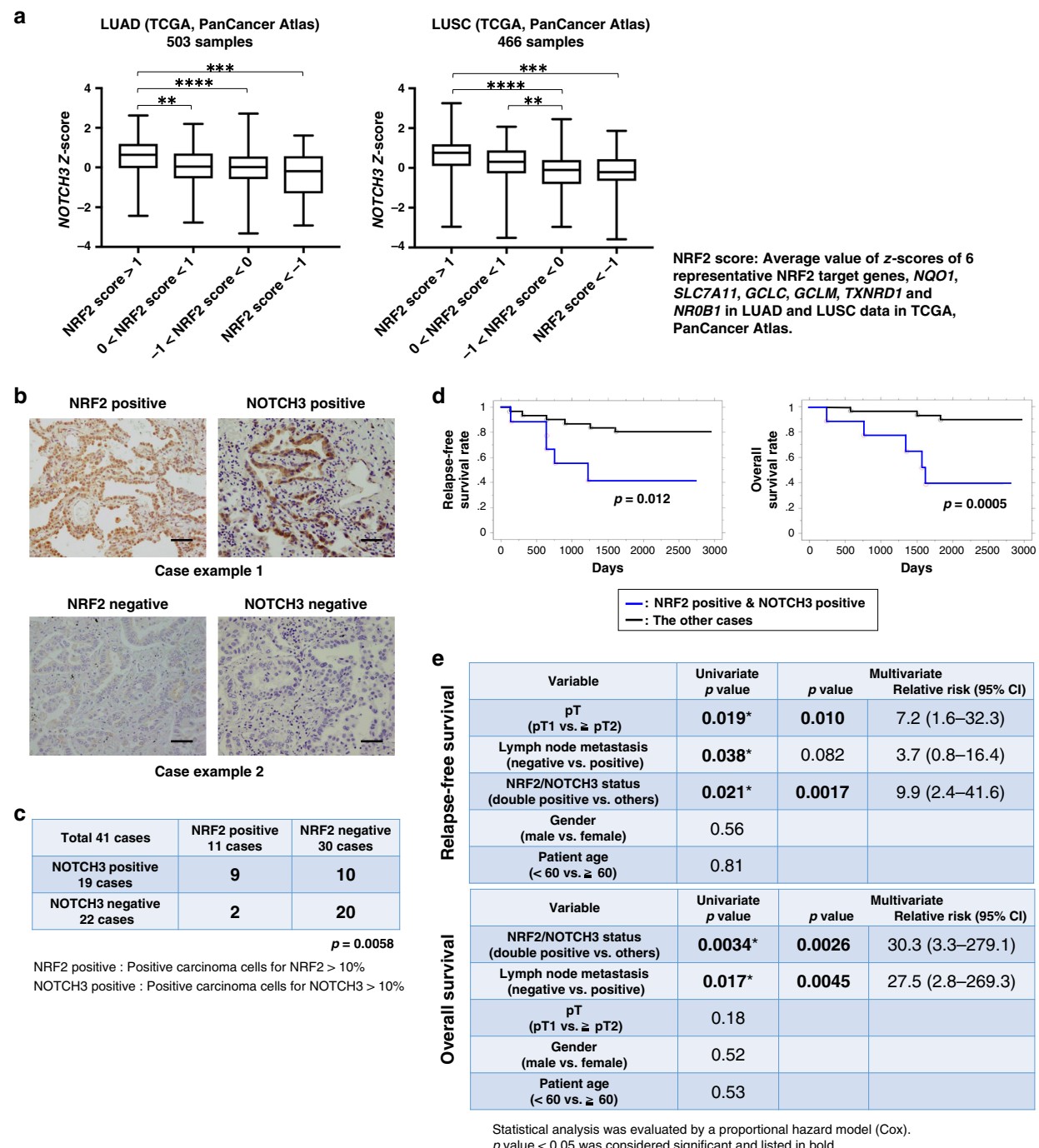

**Fig. 2 The NRF2-NOTCH3 axis is associated with poor clinical outcomes in lung adenocarcinoma patients. a** *NOTCH3* mRNA expression in tumor tissues stratified according to NRF2 activities. Gene expression data for lung adenocarcinoma (LUAD) and squamous cell carcinoma (LUSC) patients were obtained from the TCGA database. Box plots are defined by the 25th and 75th percentiles. Center line represents the median (50th percentile). Whiskers indicate minimum and maximum values. One-way ANOVA followed by the Bonferroni post hoc test was performed. \*\**p* < 0.01, \*\*\**p* < 0.001, \*\*\*\**p* < 0.0001. **b** Immunostaining for NRF2 and NOTCH3 in human lung adenocarcinoma tissues. Representative cases with high and low expressions of NRF2 and NOTCH3 from two independent experiments are shown. Scale bar correspond to 50 μm. The same cases are shown in Fig. 7c. **c** Association between NRF2 and NOTCH3 statuses in 41 lung adenocarcinoma patient samples. The Chi-square test was conducted to determine statistical significance. *p* = 0.0058. **d** Overall survival rates and relapse-free survival rates for post-surgery patients grouped into NRF2 and NOTCH3 double-positive cases and the remaining cases. Kaplan-Meier analysis was conducted. Statistical significance was evaluated using the log-rank test. *p* = 0.012 for relapse-free survival, *p* = 0.0005 for overall survival. **e** Univariate and multivariate analyses of 41 lung adenocarcinoma patients. The Cox proportional hazards model was used.

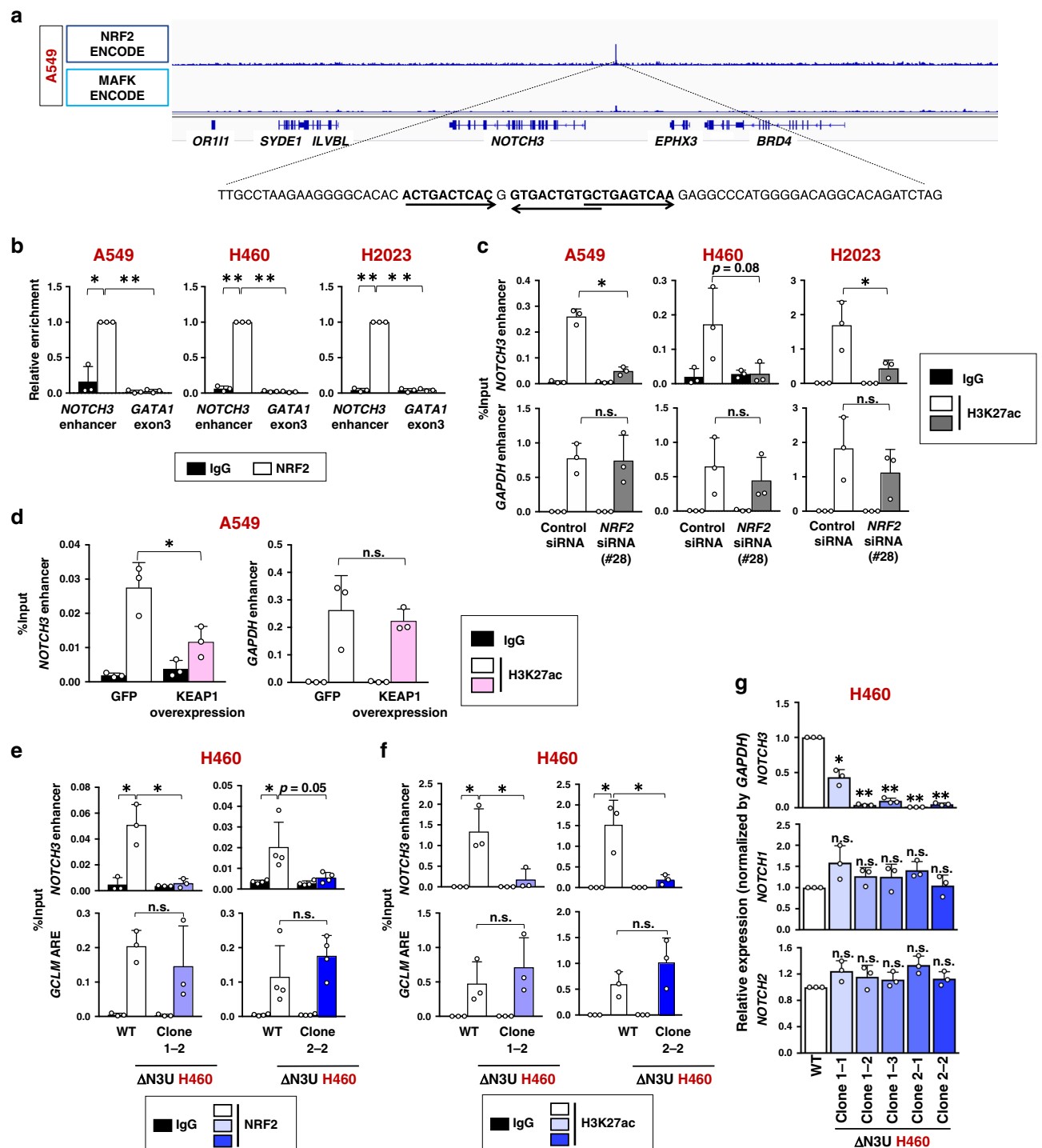

HCC4006 and H23, did not allow NRF2 binding or H3K27ac deposition at this site, whereas an apparent increase was observed in canonical NRF2 target loci (Fig. 4e, f). These results suggest that NRF2 binding and NRF2-mediated enhancer formation at the *NOTCH3* upstream region are restricted to NRF2-activated NSCLC cells.

**NRF2 generates a unique enhancer landscape in NRF2-activated NSCLCs.** The highly selective attitude of NRF2 toward the *NOTCH3* enhancer attracted our interest in the enhancer landscape of NRF2-activated NSCLCs. We examined the genome-wide distribution of NRF2 and its contribution to

enhancer formation by detecting H3K27ac deposition in A549 cells. We conducted ChIP-seq analysis of H3K27ac in A549 cells with or without *NRF2* knockdown and aligned the results with ENCODE NRF2 ChIP-seq data. NRF2 binding peaks were accompanied by H3K27ac deposition, which was reduced by *NRF2* knockdown (Supplementary Fig. 8a, b), suggesting that NRF2 generally contributes to enhancer formation. If the log2 ratio of the H3K27ac levels in *NRF2*-knockdown cells versus control cells was less than −0.5, the H3K27ac deposition was defined as NRF2-dependent (Supplementary Fig. 8c). Based on this definition, almost two-thirds of the H3K27ac deposition overlapping with NRF2-binding peaks were NRF2-dependent. A typical NRF2 binding motif was enriched in the NRF2-binding

**Fig. 3 NRF2 directly activates *NOTCH3* expression. a** ChIP-seq profile of the *NOTCH3* locus in A549 cells. Data showing NRF2 and MAFK chromatin binding were obtained from the ENCODE database. The sequence corresponding to the NRF2 binding site is shown at the bottom. Arrows indicate antioxidant response elements (AREs), which are consensus recognitions sequences of NRF2-MAFK heterodimer. **b** ChIP assay using the NRF2 antibody in NRF2-activated NSCLC cells. Enrichment of the *NOTCH3* enhancer region was examined. *GATA1* exon 3 was selected as a control locus. The average and SD of three independent experiments are shown. Two-sided confidence interval estimation was conducted to evaluate statistical significance. *$\alpha < 0.05$, **$\alpha < 0.01$. **c, d** ChIP assay using the H3K27ac antibody in three NRF2-activated NSCLC cells treated with control siRNA or *NRF2* siRNA (**c**) and A549 cells expressing GFP (control) or wild-type KEAP1 reconstitution (**d**). Enrichment of the *NOTCH3* enhancer region was examined. *GAPDH* enhancer was selected as a control locus. The average and SD of three independent experiments are shown. Two-sided Student's *t* test was performed. *$p < 0.05$, n.s.: not significant. **e, f** ChIP assay using antibodies against NRF2 (**e**) and H3K27ac (**f**) in ΔN3U and wild-type (WT) H460 cells. Enrichment of the *NOTCH3* enhancer region was examined. The *GCLM* ARE was selected as a control locus. The average and SD of three independent experiments are shown except for clone 2–2 experiment with NRF2 antibody which was independently conducted for four times. Two-sided Student's *t* test was performed. *$p < 0.05$, n.s.: not significant. **g** RT-PCR measuring the expression levels of *NOTCH1, NOTCH2,* and *NOTCH3* normalized to *GAPDH* in ΔN3U H460 cells. Fold changes of the normalized values were calculated in comparison to WT H460 cells. The average and SD of the fold changes from three independent experiments are shown. Comparison was made between two cell groups; WT H460 and each mutant clone. Two-sided confidence interval estimation was conducted for the ΔN3U H460 clones. *$\alpha < 0.05$, ** $\alpha < 0.01$, n.s.: not significant.

peaks that overlapped with NRF2-dependent H3K27ac deposition (Supplementary Fig. 8d). Of 36 NRF2-activated NSCLC-specific NRF2 downstream effector genes (see Fig. 1f and Supplementary Fig. 2a), 21 genes, including *NOTCH3*, were regarded as direct NRF2 target genes based on the presence of NRF2 binding accompanied by the NRF2-dependent H3K27ac deposition (Fig. 5a and Supplementary Fig. 8e). Similarly, 77 out of 87 canonical NRF2 downstream effector genes were regarded as direct NRF2 target genes. The 77 canonical NRF2 target genes included well-known NRF2 target genes involved in cytoprotection and metabolism. Meanwhile, the 21 NRF2-activated NSCLC-specific NRF2 target genes were associated with a variety of biological functions.

Intriguingly, the majority of the H3K27ac depositions in NRF2-activated NSCLC-specific NRF2 target loci, including the one in the *NOTCH3* locus, were not clearly detected in normal adult human lung (Fig. 5b and Supplementary Fig. 9a, b). For instance, the one at the *NR0B1* locus was strictly unique to NRF2-activated NSCLC cells (Supplementary Fig. 9b), which is consistent with a previous study describing that NR0B1 is selectively expressed in *KEAP1*-mutant NSCLC cells[38]. In contrast, H3K27ac depositions in the canonical NRF2 target gene loci were mostly observed both in A549 cells and normal adult human lung (Supplementary Fig. 9c, d). More quantitatively, H3K27ac depositions at NRF2-activated NSCLC-specific NRF2 target loci were significantly lower than those at canonical NRF2 target gene loci in normal adult human lung, whereas in A549 cells, those in both loci were similarly high (Fig. 5c).

We also compared the H3K27ac depositions in A549 and HCC4006 as representatives of NRF2-activated and NRF2-normal NSCLC cells, respectively. H3K27ac depositions in the NRF2-activated NSCLC-specific NRF2 target loci were decreased by *NRF2* knockdown in A549 cells but were not changed by DEM treatment in HCC4006 (Fig. 5d). H3K27ac depositions in the canonical NRF2 target gene loci were decreased by *NRF2* knockdown in A549 cells and increased by DEM treatment in HCC4006 cells (Fig. 5e).

Thus, H3K27 depositions at canonical NRF2 target gene loci are well detectable in normal cells as well as in NRF2-normal NSCLCs and enhanced by NRF2 activation. In contrast, those at NRF2-activated NSCLC-specific NRF2 target gene loci are observed and enhanced by NRF2 specifically in NRF2-activated NSCLCs. These results suggest that persistent NRF2 activation in cancer cells leads to the establishment of NRF2-dependent enhancers at gene loci that are not regulated by transiently activated NRF2.

**CEBPB colocalizes with NRF2 in active chromatin in NRF2-activated NSCLCs.** To clarify a molecular mechanism by which NRF2 achieves the unique enhancer formation in NRF2-activated NSCLCs, we examined the genome-wide distributions of other transcription factors in A549 cells by utilizing ENCODE ChIP-seq data sets, anticipating that transcription factor cooperativity with NRF2 would be a key to understanding the establishment of unique NRF2-dependent enhancer signatures in the special cellular context, namely, NRF2-activated NSCLCs. Among 35 transcription factors with available ChIP-seq data of A549 cells in the basal condition, top 10 transcription factors colocalizing with NRF2 were selected (Fig. 6a and Supplementary Table 1). MAFK ranked at the second place, which was reasonable as MAFK is a heterodimeric partner of NRF2. Although GATA3 and PBX3 ranked at the first and third place, respectively, their binding peaks were not robust at NRF2-dependent enhancers at 21 NRF2 target genes unique to NRF2-activated NSCLCs (data not shown). We thus chose CEBPB that ranked at the forth place for a further analysis. CEBPB colocalized with NRF2 in the active chromatin marked by H3K27ac deposition but not in the region without H3K27ac deposition (Fig. 6b), implying a functional interaction between NRF2 and CEBPB. As expected, CEBPB was detected in the endogenous NRF2 transcription complex in A549 cells (Fig. 6c). Notably, CEBPB protein levels in NRF2-activated NSCLC cells were higher than those in NRF2-normal NSCLC cells (Fig. 6d). KEAP1 reconstitution in NRF2-activated NSCLC cells, which canceled NRF2 accumulation, reduced CEBPB protein levels (Fig. 6e). These results suggest that a sufficient availability of CEBPB is one of the critical factors that invigorates NRF2 for the enhancer remodeling in NRF2-activated NSCLCs. Three out of six NRF2-dependent enhancers unique to NRF2-activated NSCLCs, those in *FAM20B, ZC3H12A,* and *C5AR1* loci, exhibited reduced H3K27ac deposition by *CEBPB* knockdown (Fig. 6f), suggesting that cooperativity between NRF2 and CEBPB explains a part of unique enhancer formation in NRF2-activated NSCLCs.

The NRF2-CEBPB cooperativity was further evaluated in clinical samples. Positive correlation was observed between NRF2 and CEBPB protein levels in the surgically resected LUAD tumors (Fig. 7a, b). Immunohistochemical detection of CEBPB in the 41 LUAD samples, which were examined for NRF2 and NOTCH3 expression (see Fig. 2b), showed statistically significant association between CEBPB and NRF2 (Fig. 7c, d). Strongly NRF2-positive cases tended to be positive for CEBPB and NOTH3 (Fig. 7e), and out of 7 NRF2-CEBPB double positive cases, 6 were NOTCH3 positive (Fig. 7f), supporting our observation in cell lines that abundantly accumulated NRF2 activates *NOTCH3* with the aid of CEBPB in NRF2-activated NSCLCs.

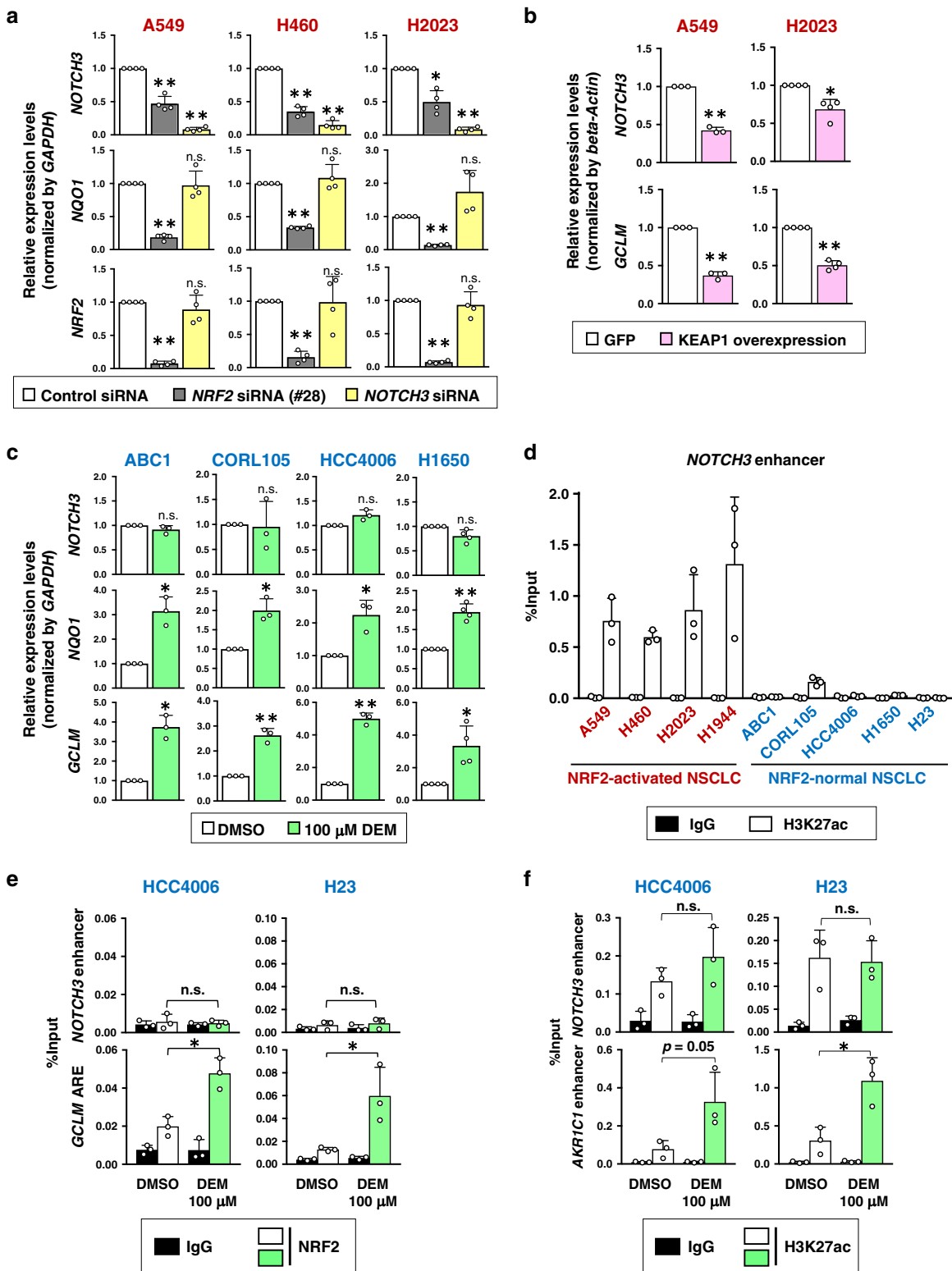

**Cooperative chromatin binding of NRF2 and CEBPB generates NOTCH3 enhancer in NRF2-activated NSCLCs.** The *NOTCH3* enhancer region was found to be bound by seven transcription factors, including CEBPB, in A549 cells according to the ENCODE ChIP-seq data (Fig. 8a) and contained consensus binding sites for eight transcription factors, among which those for NRF2/FOSL2 and CEBPB were conserved in the human and mouse (Fig. 8b). Knockdown experiments in A549 cells revealed that CEBPB and FOSL2 made a substantial

contribution to the *NOTCH3* expression (Fig. 8c and Supplementary Fig. 10a).

At the *NOTCH3* enhancer region, *FOSL2* knockdown decreased H3K27ac deposition but did not affect NRF2 binding (Supplementary Fig. 10b, c), and *NRF2* knockdown did not alter FOSL2 binding (Supplementary Fig. 10d), indicating that FOSL2 contributes to *NOTCH3* enhancer formation independently of NRF2. In contrast, *CEBPB* knockdown decreased both H3K27ac deposition and NRF2 binding (Fig. 8d, e), and *NRF2* knockdown

**Fig. 4 The *NOTCH3* enhancer is uniquely formed in NRF2-activated NSCLCs. a**, **b** RT-PCR measuring the expression of *NOTCH3*, *NQO1*, and *NRF2* normalized to *GAPDH* in three NRF2-activated NSCLC cells treated with control siRNA, *NRF2* siRNA or *NOTCH3* siRNA (**a**) and the expression of *NOTCH3* and *GCLM* normalized to *beta-Actin* in two NRF2-activated NSCLC cells with GFP expression (control) or wild-type KEAP1 reconstitution (**b**). Fold changes of the normalized values were calculated in comparison to control siRNA treatment. The average and SD of the fold changes from four independent experiments are shown except for the experiment with A549 in **b**, which was independently conducted for three times. Comparison was made between two cell groups; control siRNA and each test siRNA (**a**). Two-sided confidence interval estimation was conducted to evaluate statistical significance. *$\alpha <$ 0.05, **$\alpha <$ 0.01, n.s.: not significant. **c** RT-PCR measuring the expression of *NOTCH3*, *NQO1*, and *GCLM* normalized to *GAPDH* in four NRF2-normal NSCLC cells treated with vehicle (DMSO) or DEM. Fold changes of normalized values were calculated in comparison to vehicle-treated samples. The average and SD of the fold changes from three independent experiments are shown except for the experiment with H1650, which was independently conducted for four times. Two-sided confidence interval estimation was conducted to evaluate statistical significance. *$\alpha <$ 0.05, **$\alpha <$ 0.01, n.s.: not significant. **d** ChIP assay using the H3K27ac antibody in NRF2-activated and NRF2-normal NSCLC cells. The average and SD of three independent experiments are shown. **e**, **f** ChIP assay using the NRF2 (**e**) or H3K27ac (**f**) antibodies in two NRF2-normal NSCLC cells treated with vehicle (DMSO) or DEM. Enrichment of the *NOTCH3* enhancer region was examined. The *GCLM* ARE (**e**) and *AKR1C1* enhancer (**f**) were selected as positive control loci. The average and SD of three independent experiments are shown. Two-sided Student's *t* test was performed. *$p < 0.05$, n.s.: not significant. DMSO: dimethyl sulfoxide, DEM: diethylmaleate.

decreased CEBPB binding (Fig. 8f), suggesting the cooperative binding of NRF2 and CEBPB to the *NOTCH3* enhancer. We further verified the necessity of NRF2 for CEBPB binding to the *NOTCH3* enhancer by utilizing ΔN3U A549 cells, which harbor deletions in NRF2 binding sites (Fig. 8g). In spite of the presence of an intact CEBPB binding motif, CEBPB binding to the *NOTCH3* enhancer was decreased in both clones of ΔN3U A549 cells (Fig. 8h), verifying the cooperative binding of NRF2 and CEBPB.

A significant contribution of CEBPB to *NOTCH3* expression was also observed in other NRF2-activated NSCLC cells, H460 and H2023 (Supplementary Fig. 10e, f), suggesting that NRF2-CEBPB cooperation is commonly important in NRF2-activated NSCLC cells. Thus, the *NOTCH3* enhancer comprises a unique enhancer landscape of NRF2-activated NSCLCs, which is shaped by NRF2-CEBPB cooperativity.

**The NOTCH3 enhancer is critical for the tumor-initiating activity of NRF2-activated NSCLCs.** Finally, we examined the impact of the NRF2-NOTCH3 axis on the malignant behavior of NRF2-activated NSCLCs, which was suggested from the clinical study, in terms of tumor-initiating activity. Abrogation of the NRF2-NOTCH3 axis by disrupting the *NOTCH3* enhancer suppressed oncosphere growth of NRF2-activated NSCLC cell lines (Fig. 9a and Supplementary Fig. 11a, b). Knockdown of *CEBPB*, which was found to be a key cooperative factor with NRF2 for the *NOTCH3* enhancer formation, similarly suppressed the oncosphere growth of NRF2-activated NSCLC cell lines (Supplementary Fig. 11c, d). Of note, disruption of the *NOTCH3* enhancer hardly affected spheroid growth (Fig. 9b), which reflects a simple cell growth ability. These results are in a good contrast with those of *NRF2* knockdown in NRF2-activated cancers. *NRF2* knockdown impaired both oncosphere growth and spheroid growth (Supplementary Fig. 11e and see Fig. 1a–c), indicating that NRF2 promotes cell proliferation and survival in addition to tumor-initiating activity. Thus, the NRF2-NOTCH3 axis specifically contributes to tumor-initiating activity, which should be distinguished from other functional axes driven by NRF2 for malignant progression of cancers, such as cell proliferation and survival.

We next conducted xenograft experiments to examine the in vivo contribution of the *NOTCH3* enhancer to the promotion of tumor-initiating activity in NRF2-activated NSCLC cells. The tumor growth of *NOTCH3* enhancer-disrupted cells, ΔN3U H460, ΔN3U A549, and ΔN3U H2023, was decreased compared with that of their parental cells (Fig. 9c and Supplementary Fig. 12). The suppressed tumor growth was restored by supplementation with the intracellular domain of NOTCH3

(N3ICD) (Fig. 9d and Supplementary Fig. 13a–c), supporting the notion that the *NOTCH3* enhancer mediates the effects of hyperactivated NRF2 on tumor growth. To further verify that the *NOTCH3* enhancer promotes the tumor-initiating activity of NRF2-activated NSCLC cells, we conducted a serial transplantation assay and compared the frequency of tumorigenesis (Fig. 9e and Supplementary Fig. 13d, e). After the secondary transplantation, both clones of ΔN3U H460 cells generated a reduced number of tumors compared with parental H460 cells, supporting the notion that the NRF2-NOTCH3 axis contributes to the improved maintenance of tumor-initiating activity. Expecting therapeutic efficacy improvement, we combined *NOTCH3* enhancer inhibition targeting TICs and cytotoxic anti-cancer drugs, such as cis-dichlorodiammineplatinum (CDDP), targeting the proliferating population. The tumorigenesis of ΔN3U H460 cells was suppressed by CDDP more effectively than that of parental cells (Fig. 9f), successfully demonstrating a synergistic effect of CDDP and *NOTCH3* enhancer inhibition.

## Discussion

We linked persistent NRF2 activation in NSCLC cells to unique enhancer formation at the *NOTCH3* locus and demonstrated the clinical relevance of the NRF2-NOTCH3 regulatory axis. The significant anti-tumorigenic effect caused by the disruption of a single enhancer, i.e., disruption of the *NOTCH3* enhancer, stresses that enhancer dysregulation plays a critical role in the biological characteristics of cancers. The enhancer remodeling that occurs in NRF2-activated NSCLCs is partly mediated by a unique NRF2 transcription complex containing CEBPB. The accumulation of CEBPB is likely to modulate the NRF2 cistrome, which confers increased tumor-initiating activity on NRF2-activated NSCLCs by establishing the NRF2-NOTCH3 axis (Fig. 10).

NRF2 is capable of both inhibiting and promoting carcinogenesis depending on the cellular context[39]. It is currently unclear both when and how NRF2 switches between its role as a guardian of cells that maintains redox homeostasis and its role as a driver of cancers that enhances aggressive tumorigenesis and therapeutic resistance. Based on studies demonstrating that sole overexpression of NRF2 in normal cells does not cause carcinogenesis, it is clear that constitutive activation of NRF2 by *KEAP1* or *NRF2* mutations, which are often encountered in NRF2-activated cancers, is not by itself cancer driver[40–42]. We surmise that the enhancer remodeling is one of the requirements for the establishment of NRF2-activated cancers exhibiting NRF2 addiction. Although previous studies described that enhancer remodeling by a key transcription factor forms a fundamental basis for malignant behaviors of cancer cells[43,44], mechanistic distinction in

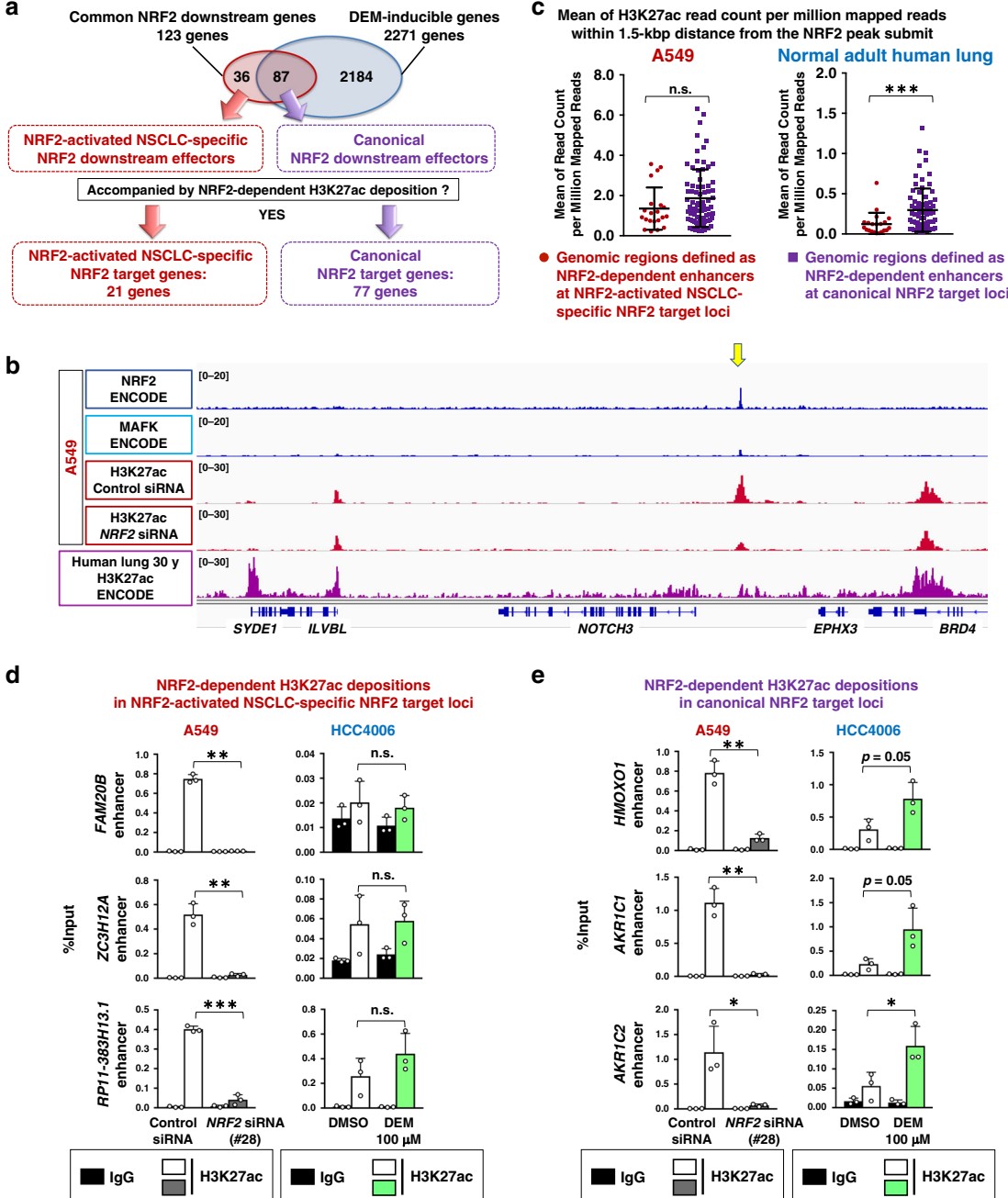

**Fig. 5 NRF2-activated NSCLC possesses unique enhancer signatures. a** NRF2-activated cancer-specific NRF2 target genes and canonical NRF2 target genes based on the presence of NRF2-dependent enhancers. **b** ChIP-seq profiles at the *NOTCH3* locus. A549 cells (upper panels) and normal adult human lung sample (lower panel) are shown. NRF2 and MAFK chromatin binding in A549 cells and H3K27ac deposition pattern in normal human lung sample were obtained from the ENCODE database. Acetylated H3K27 deposition profiles in A549 cells treated with control siRNA or *NRF2* siRNA were obtained in this study. A yellow arrow indicates NRF2-dependent H3K27ac deposition. **c** Comparison of H3K27ac deposition in NRF2-dependent enhancers that were defined in A549 cells. The NRF2-dependent enhancers were classified into two groups: those within NRF2-activated NSCLC-specific NRF2 target loci and those within canonical NRF2 target loci. H3K27ac deposition in A549 cells was compared between these two groups using merged results of three independent experiments (left panel). Similar comparison was conducted using normal adult human lung samples in ENCODE database (right panel). Two-tailed Mann–Whitney test was performed. ***$p < 0.0005$, n.s.: not significant. **d**, **e** ChIP assay using the H3K27ac antibody in A549 cells treated with control or *NRF2* siRNA and HCC4006 cells treated with vehicle (DMSO) or DEM. H3K27ac deposition at NRF2-activated NSCLC-specific NRF2 target loci (**d**) and canonical NRF2 target loci (**e**) were examined. The average and SD of three independent experiments are shown. Two-sided Student's *t* test was performed. *$p < 0.05$, **$p < 0.005$, ***$p < 0.0005$, n.s.: not significant. DMSO: dimethyl sulfoxide, DEM: diethylmaleate.

roles of the transcription factor has not been well understood under physiological conditions and in the context where it serves as a cancer driver. In this study, we found that CEBPB is one of the factors that alter NRF2 function in NRF2-activated NSCLC; CEBPB is involved in the enhancer remodeling that invigorates

NRF2 for transcriptional activation of non-canonical target genes. Based on the close proximity between AREs and a CEBPB binding motif in the *NOTCH3* enhancer region, NRF2 and CEBPB appear to bind to DNA as distinct dimers with MAFK (or other small MAF) and C/EBP family members, respectively.

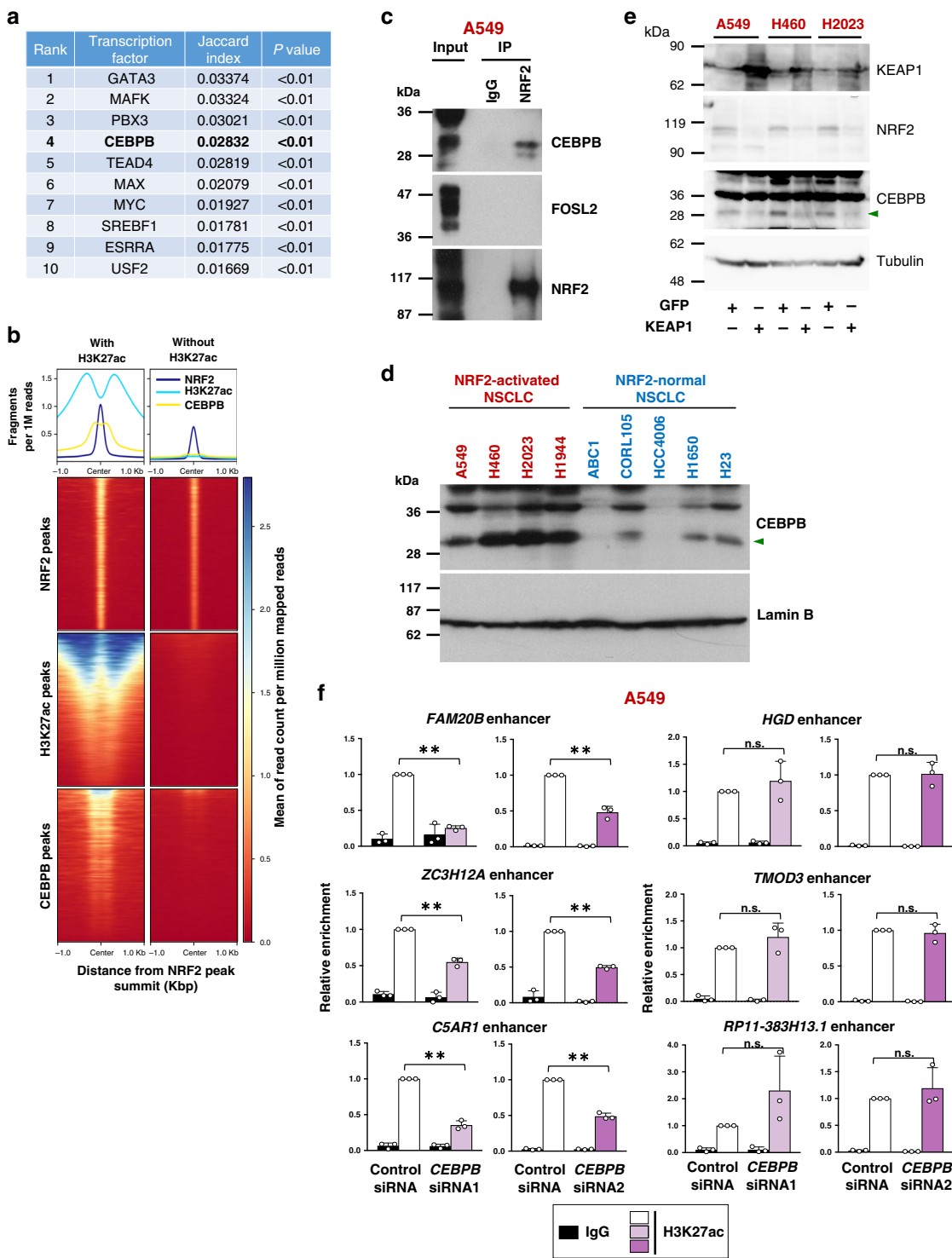

**Fig. 6 NRF2 cooperates with CEBPB for enhancer formation in NRF2-activated NSCLCs. a** Genome-wide colocalization of NRF2 and other transcription factors at active enhancers marked with H3K27ac deposition in A549 cells. Colocalization tendency was analyzed by Jaccard test using ChIP-seq data deposited in ENCODE. Higher rank indicates better colocalization with NRF2. **b** Aggregation plots and heat maps of ChIP-seq data of NRF2, H3K27ac, and CEBPB surrounding NRF2 binding sites obtained from the ENCODE database. **c** Affinity purification of the NRF2 complex from A549 cells. A representative result of two independent experiments is shown. **d** Immunoblot analysis of CEBPB protein levels in nuclear extracts of four NRF2-activated and five NRF2-normal NSCLC cells. An arrowhead indicates the CEBPB isoform that corresponds to the one detected in the immunoprecipitation experiment with the NRF2 antibody shown in **c**. Lamin B expression was used as a loading control. The results shown are representative of 2 independent experiments. **e** Immunoblot analysis detecting KEAP1, NRF2, and CEBPB in three NRF2-activated NSCLC cells with GFP expression (control) or wild-type KEAP1 reconstitution. Tubulin expression was used as a loading control. The results shown are representative of three independent experiments. **f** ChIP assay using the H3K27ac antibody in A549 cells treated with control or *CEBPB* siRNA. H3K27ac deposition at NRF2-activated NSCLC-specific NRF2 target loci were examined. The average and SD of three independent experiments are shown. Two-sided confidence interval estimation was conducted for knockdown samples incubated with H3K27ac antibody. **$\alpha$ < 0.01, n.s.: not significant.

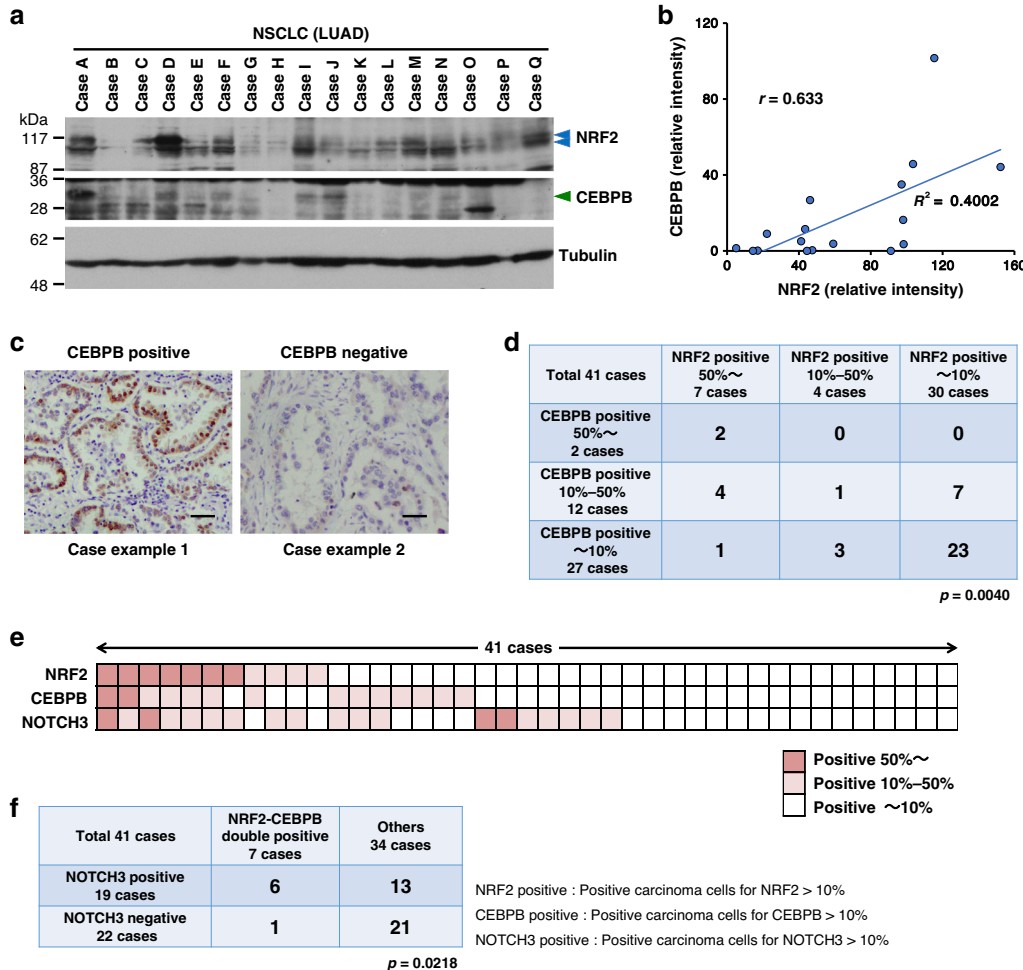

**Fig. 7 CEBPB and NRF2 are frequently co-expressed in tumor tissues of lung adenocarcinoma. a** Immunoblot analysis of NRF2 and CEBPB in primary LUAD tissues. Blue arrowheads indicate specific bands of NRF2. A green arrowhead indicates the CEBPB isoform that corresponds to the one detected in the immunoprecipitation experiment with the NRF2 antibody shown in Fig. 6c. Tubulin was detected as a loading control. A representative result from two independent experiments is shown. LUAD; lung adenocarcinoma. **b**. Plots of band intensities of NRF2 and CEBPB shown in **a**. r: Pearson's correlation coefficient, $R^2$: R-Squared, p: two-tailed p value. **c** Immunostaining for CEBPB in LUAD tissues. Representative cases with high and low expressions of CEBPB from two independent experiments are shown. Scale bars correspond to 50 μm. The same cases are shown in Fig. 2b. **d**–**f** Association of NRF2 and CEBPB statuses (**d**), a heatmap of immunoreactivities of NRF2, CEBPB and NOTCH3 (**e**) and association of NRF2/CEBPB and NOTCH3 statuses (**f**) of the 41 lung adenocarcinoma cases. The Chi-square test was conducted to determine statistical significance (**d**, **f**). p = 0.0040 (**d**), p = 0.0218 (**f**).

Although precise mechanisms for the cooperative binding of these dimers are unknown, considering previous reports describing that CEBPB serves as a pioneer factor for enhancer establishment in cancer cells as well as in normal hematopoietic stem cells[45,46], CEBPB may contribute to create a permissive chromatin environment for NRF2-small Maf heterodimer to access regulatory regions of unique target genes in NRF2-activated NSCLC cells.

Our clinical study demonstrated that NRF2-NOTCH3 double-positive patients tended to have a significantly poorer prognosis compared with the others. These results suggest that not only NOTCH3 but other NRF2 canonical targets such as cytoprotective genes would also contribute to the poorer prognosis in NRF2-activated NSCLC. An important consideration here is that the promotion of tumor-initiating activity in TICs must be coupled with enhanced proliferation and survival of differentiated cancer cells in order to make a substantial contribution to cancer malignancy.

As a therapeutic perspective, NRF2 would be the best target for eliminating NRF2-activated cancers. Indeed, enthusiastic efforts are being made by many laboratories and pharmaceutical companies to develop NRF2 inhibitors for cancer cases that exhibit abnormal NRF2 activation and are therefore refractory to normal chemo- and radiotherapies[47,48]. However, considering the critical protective functions of NRF2, systemic administration of NRF2 inhibitors may cause deleterious effects in cancer-bearing hosts[49,50]. In this study, we explored unique NRF2 target genes and identified NOTCH3 as a key regulator of the tumor-initiating activity in NRF2-activated NSCLCs. Because Notch3-deficient mice exhibit modest vascular phenotypes but no serious defects[51,52] and because NOTCH3 is not involved in NRF2-mediated cytoprotection, NOTCH3 inhibition is expected to exert anti-cancer effects without interfering with normal cellular functions in cancer-bearing hosts. As NRF2-activated cancer cells are capable of highly efficient drug extrusion, targeting NOTCH3 is further advantageous in that it enables the circumvention of NRF2-mediated chemo-resistance. This is because NOTCH3 can be antagonized from the outside of cells at its functionally important extracellular domain. NOTCH3 inhibition is expected to efficiently reduce the recurrence of NRF2-activated cancers by suppressing tumor-initiating activities without having adverse effects on cancer-bearing hosts.

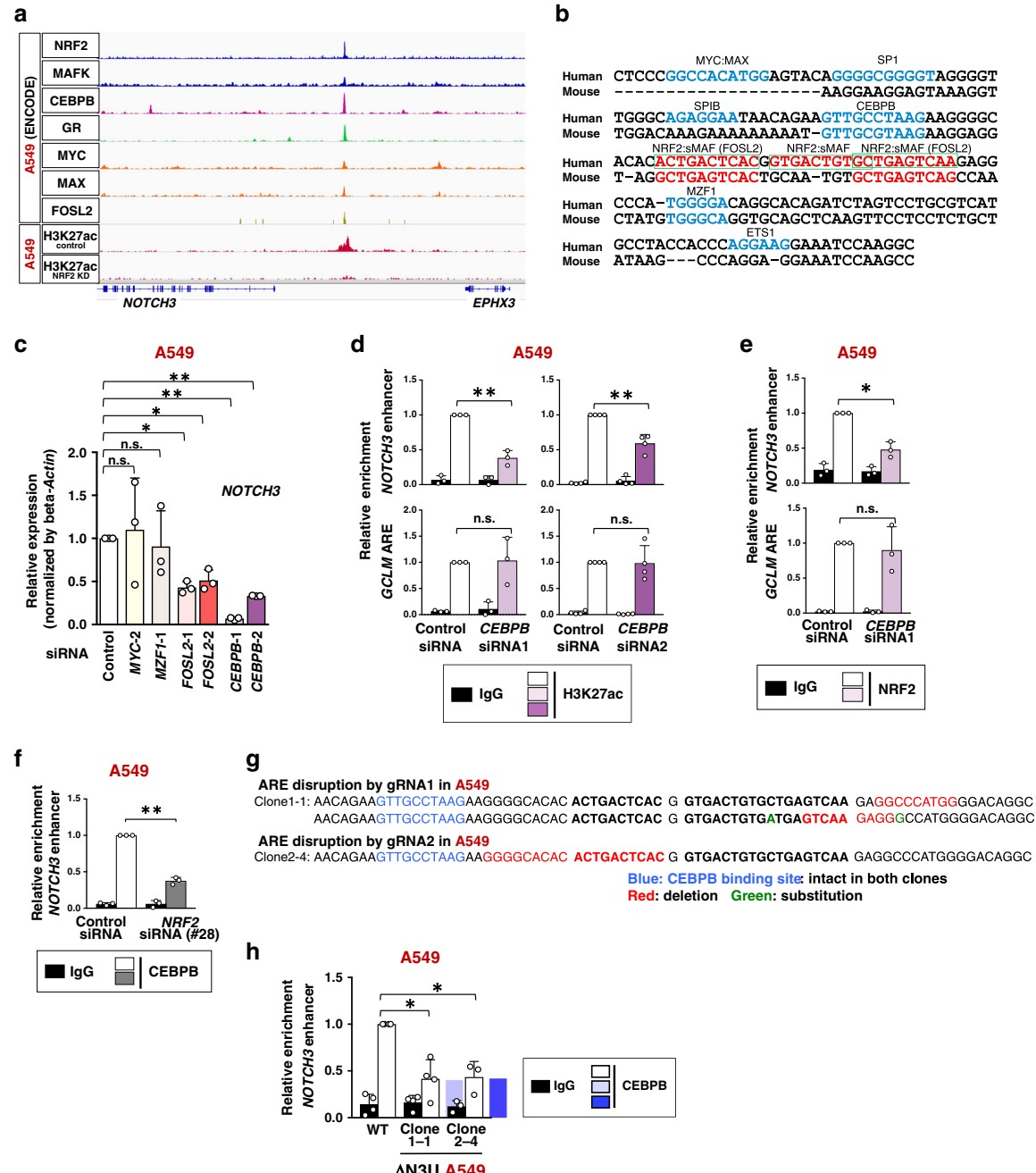

**Fig. 8 CEBPB-NRF2 cooperation is required for *NOTCH3* enhancer formation in NRF2-activated NSCLC cells. a** ChIP-seq profiles of the *NOTCH3* locus in A549 cells. **b** Sequence comparison of flanking regions surrounding NRF2 binding sites in the *NOTCH3* enhancer between human and mouse. AREs (NRF2: sMAF binding sites) are shown in red, and three tandem human AREs are enclosed by green frames. Consensus binding motifs of other transcription factors are shown in blue. **c** RT-PCR of *NOTCH3* in A549 cells treated with siRNAs. Fold changes of the normalized values were calculated in comparison to A549 cells treated with control siRNA. The average and SD of the fold changes from three independent experiments are shown. Comparison was made between two cell groups; control siRNA and each test siRNA. Two-sided confidence interval estimation was conducted to evaluate statistical significance. $*\alpha < 0.05$, $**\alpha < 0.01$, n.s.: not significant. **d**, **e** ChIP assay using the H3K27ac (**d**) and NRF2 (**e**) antibodies in A549 cells treated with control siRNA or *CEBPB* siRNA. Fold changes of %input values were calculated in comparison to the control samples incubated with H3K27ac or NRF2 antibody. The average and SD of three independent experiments are shown except for the experiment with *CEBPB* siRNA2 in **d**, which was independently conducted for four times. Two-sided confidence interval estimation was conducted for knockdown samples incubated with H3K27ac or NRF2 antibody. $*\alpha < 0.05$, $**\alpha < 0.01$, n.s.: not significant. **f** ChIP assay using the CEBPB antibody in A549 cells. Fold changes of %input values were calculated in comparison to the control samples reacted with CEBPB antibody. The average and SD of 3 independent experiments are shown. Two-sided confidence interval estimation was conducted for knockdown samples reacted with CEBPB antibody. $**\alpha < 0.01$. **g** DNA sequence of ΔN3U A549 cells. A CEBPB binding site, shown in blue, is preserved in both ΔN3U A549 clones. Deletion and substitution by genome editing are shown in red and green, respectively. **h** ChIP assay using the CEBPB antibody in ΔN3U and wild-type (WT) A549 cells. Fold changes of %input values were calculated in comparison to WT A549 cells incubated with CEBPB antibody. The average and SD of 3 (Clone 2–4) and 4 (WT and Clone 1–1) independent experiments are shown. Comparison was made between two cell groups; WT A549 and each mutant clone. Two-sided confidence interval estimation was conducted for Clones 1–1 and 2–4 incubated with CEBPB antibody. $*\alpha < 0.05$.

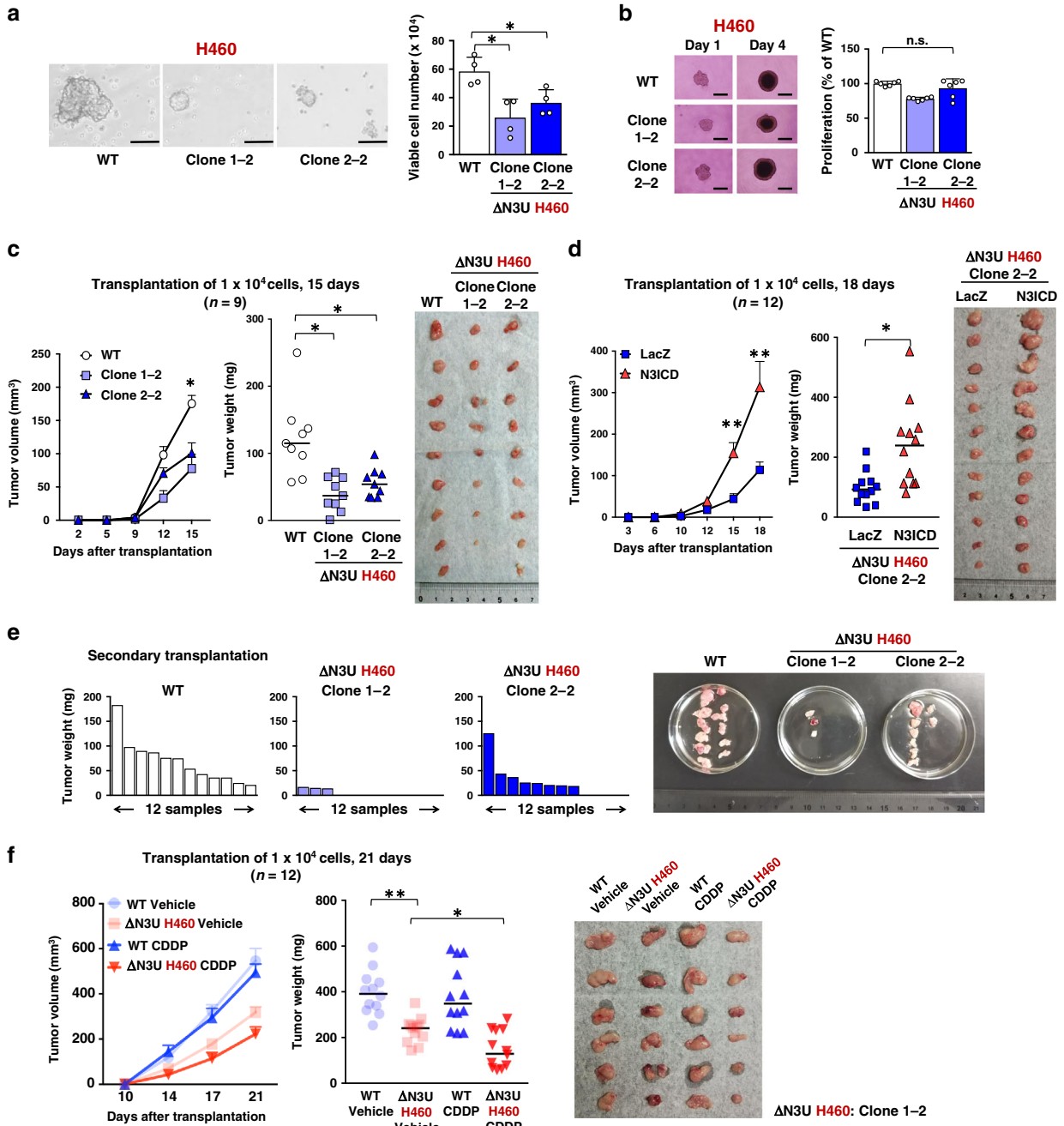

**Fig. 9 The *NOTCH3* enhancer promotes tumor-initiating activity of NRF2-activated NSCLCs. a** Oncosphere growth of ΔN3U and wild-type (WT) H460 cells (left panel). Scale bars indicate 100 μm. Viable cells were counted after trypsinization (right panel). Average cell numbers and SD from four independent experiments are shown. Comparison was made between two cell groups; WT and each mutant clone. Two-sided Student's *t* test was performed. \**p* < 0.05. **b** Spheroid growth of ΔN3U and WT H460 cells (left panels). Scale bars indicate 100 μm. Cell numbers were estimated using a cell counting kit on day 4 (right panel). Average cell numbers and SD from six independent experiments are shown. The average number of WT H460 cells was set as 100%. Two-sided Student's *t* test was performed. n.s.: not significant. **c** Xenograft experiment of ΔN3U and WT H460 cells (*n* = 9 each; number of xenograft tumors). A photograph shows xenograft tumors at the time of tumor weight measurement on day 15. Horizontal bars indicate the median tumor weight (middle panel). Data are presented as mean + SEM (left panel). Two-sided Wilcoxon rank sum test was performed. \**p* < 0.05. **d** Xenograft experiment of ΔN3U H460 Clone 2–2 cells with LacZ and N3ICD (*n* = 12 each; number of xenograft tumors). A photograph shows xenograft tumors at the time of tumor weight measurement on day 18. Horizontal bars indicate the median tumor weight (middle panel). Data are presented as mean + SEM (left panel). Two-sided Wilcoxon rank sum test was performed. \**p* < 0.05, \*\**p* < 0.005. **e** Serial transplantation experiment of ΔN3U and WT H460 cells. Each graph shows the weight of tumors in the secondary transplantation. A photograph shows tumors at the time of weight measurement in the secondary transplantation. **f** CDDP treatment experiment into nude mice after transplantation of ΔN3U and WT H460 cells. A representative photograph shows xenograft tumors at the time of tumor weight measurement on day 21. Horizontal bars indicate the median tumor weight (middle panel). Data are presented as mean + SEM (left panel). One-way ANOVA followed by the Bonferroni post hoc test was performed. \**p* < 0.05, \*\**p* < 0.005. *n* = 12 biologically independent samples in each four group.

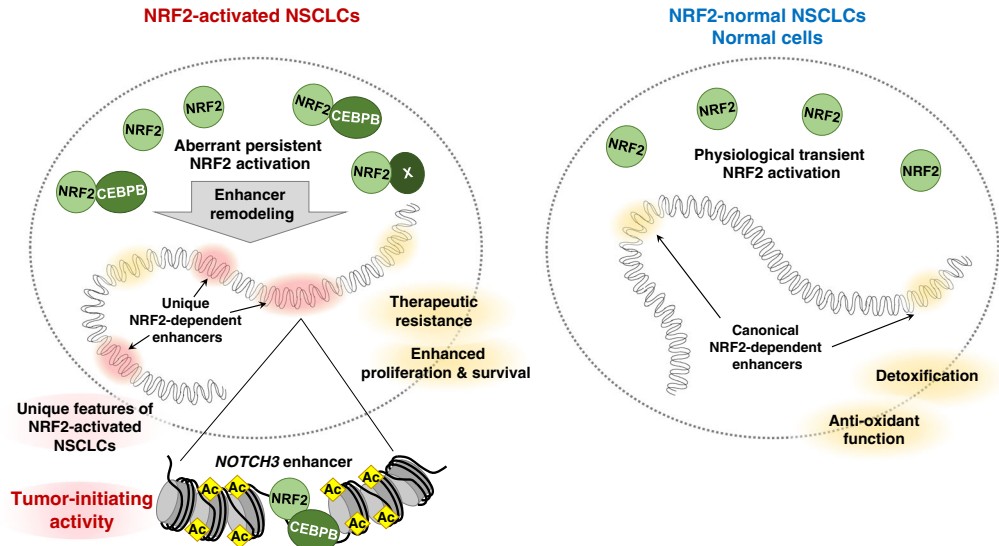

**Fig. 10 Illustration of enhancer remodeling in NRF2-activated NSCLCs.** Under physiological conditions and in NRF2-normal NSCLCs, NRF2 is transiently activated in response to stimuli and induces canonical enhancer formation to activate genes involved in detoxification and antioxidant function (right). The canonical enhancers are regarded to confer therapeutic resistance and promote cell proliferation and survival. In NRF2-activated NSCLCs, enhancer remodeling creates unique NRF2-dependent enhancers (left). CEBPB is one of the cooperative factors for NRF2 that mediates enhancer remodeling. The *NOTCH3* enhancer, one of the unique enhancers, promotes tumor-initiating activity.

Enhancer signatures have been shown to be more dynamic according to cell types and tissue types than those of promoters[53–55], which indicates that the same gene expressed in different cell types is often regulated by the same promoter but by different enhancers. Cancer-specific enhancers are expected to be ideal therapeutic targets because their inhibition would not interfere with the expression of respective genes in normal tissues. Modified nucleic acids and nucleic acid mimetics are possible candidates enabling the control of a specific enhancer by invading the DNA double strand and forming a triplex at the enhancer region of interest[56]. Clarification of enhancer signatures responsible for the cancer malignancy and intervention to interfere with enhancer activities is an ambitious future challenge in development of anti-cancer drugs.

## Methods

**Cell culture**. NRF2-activated NSCLC cell lines (A549, NCIH460 and NCIH2023 and NCIH1944) and NRF2-normal NSCLC cell lines (ABC1, CORL105, HCC4006, NCIH1650, and NCIH23) were used in this study. A549 cells were maintained in high glucose DMEM supplemented with 10% fetal bovine serum (FBS) and penicillin/streptomycin (Gibco). The rest of the cells were maintained in low glucose DMEM supplemented with 10% FBS and penicillin/streptomycin (Gibco). The cells were maintained in a 5% $CO_2$ atmosphere at 37 °C. PCR was used to confirm that the cultured cells were not infected with mycoplasma.

**Patients and tissue specimens**. Tumor tissue specimens for histological analysis were obtained from randomly selected 41 lung adenocarcinoma patients who underwent surgical resection without preoperative treatments, irradiation or chemotherapy, between 2003 and 2004 in the Department of Thoracic Surgery at Tohoku University Hospital. The mean patient age was 66 years (range 37–82 years), with the exception of one case whose age was unknown. The mean follow-up period was 1982 days (range 233-2949). All specimens were fixed in 10% formalin and embedded in paraffin wax.

Tumor tissue specimens for immunoblot analysis were obtained from 17 lung adenocarcinoma patients who underwent surgical resection during 2019 in the Department of Thoracic Surgery at Tohoku University Hospital. All specimens were snap-frozen in liquid nitrogen right after the resection. All research protocols involved in this study were approved by the Ethics Committees at Tohoku University Graduate School of Medicine, and informed consent was obtained from the participants.

**Mice**. Four-week-old male BALB/cAJcl-nu/nu mice (CLEA Japan) and 8–12 week-old male *Keap1*-knockdown (*Keap1* KD)[57] and their control wild-type mice were used in this study. All animals were housed in air-conditioned room at an ambient

temperature of 20–26 °C, humidity of 30–70% and 12-h dark/light cycle. They were housed in specific pathogen-free conditions, according to the regulations of The Standards for Human Care and Use of Laboratory Animals of Tohoku University and the Guidelines for Proper Conduct of Animal Experiments by the Ministry of Education, Culture, Sports, Science, and Technology of Japan.

**Immunoblot analyses**. For the preparation of nuclear lysates, cells were lysed in buffer A (10 mM HCl (pH7.5) 10 mM KCl, 1.5 mM $MgCl_2$, 0.1% NP40), and crude nuclei were pelleted by centrifugation and lysed in 2x Laemmli buffer followed by boiling at 95 °C for 5 min. For the preparation of whole-cell lysates, cells were directly lysed in 2x Laemmli buffer followed by boiling at 100 °C for 10 min. The protein samples were separated by SDS-PAGE and transferred onto PVDF membranes (Immobilon P, Millipore). The antibodies used are as follows: anti-NRF2 (sc-13032X, Santa Cruz; 1:1,000-1:500), anti-KEAP1 (#111; 1:200)[58], anti-NOTCH3 (ab23426, Abcam; 1:2,000), anti-CEBPB antibody (sc-150 X, Santa Cruz; 1:1,000), anti-Tubulin (T9026, Sigma; 1:5,000-1:2,000) and anti-Lamin B (sc-6217, Santa Cruz; 1:2,000). Quantification of band intensities in the immunoblot data of human samples were conducted by ImageJ software (ver. 1.45 s; http://imagej.nih.gov/ij). Pearson's correlation coefficient was calculated for correlation of band intensities between NRF2 and CEBPB.

**Transient knockdown experiments**. NRF2 siRNAs were purchased from Invitrogen (cat. no. HSS107128 and HSS107130). *NOTCH3* siRNA was purchased from Dharmacon (siGENOME SMARTpool siRNA D-011093). ON-TARGETplus SMARTpool siRNAs were purchased for *F2RL2* (L-005491), *FAM20B* (L-021203), *GPI* (L-004900), *GSTP1* (L-011179), *HGD* (L-009823), *HJURP* (L-015443), *HMGA1* (L-004597), *HPDL* (L-014985), *NRG4* (L-015692), *PFN2* (L-011750), *SRPK1* (L-003982) and *VRK1* (L-004683) from Dharmacon. *CEBPB* siRNAs (SASI_Hs01_00236022 and SASI_Hs01_00236027), *FOSL2* siRNAs (SASI_Hs01_00057657 and SASI_Hs02_00339278), *MYC* siRNA (SASI_Hs01_00222676) and *MZF1* siRNA (SASI_Hs01_00096728) were purchased from Sigma-Aldrich. MISSION siRNA Universal Negative Controls (Sigma-Aldrich) or DS scrambled negative control siRNA (IDT) were used as controls. siRNAs were transfected into cells either by electroporation using an MP-100 MicroPorator (Digital Bio Technology) or by lipofection using Lipofectamine™RNAiMAX Transfection Reagent (Thermo Fisher Scientific). Culture media were changed 24 hrs after transfection. After another 24-48 hrs, the cells were harvested for RNA purification, immunoblot analysis and the ChIP assay. The protocols used for transient knockdown with spheroid formation assay and oncosphere formation assays in each cell line are described below.

**Transient induction of NRF2**. Cells were treated with 100 μM of diethyl maleate (DEM) or 30 nM of CDDO-Im and dimethyl sulfoxide (DMSO) as a vehicle. Cells were harvested either 4 h after treatment for immunoblot analysis and the ChIP assay or 16 h after treatment for RNA purification.

**RNA purification and RT-PCR.** Total RNA was prepared from cells using ISO-GEN (NIPPON GENE). cDNA was synthesized from 0.5 µg of total RNA using ReverTra Ace qPCR RT Master Mix with gDNA Remover (TOYOBO). Quantitative real-time PCR was performed for each sample in duplicate with Probe qPCR Mix (TOYOBO), SYBR qPCR Mix (TOYOBO) or PowerUp SYBR Green Master Mix (Thermo Fisher Scientific) using the Applied Biosystems 7300 PCR system (Applied Biosystems) and Applied Biosystems Quant Studio 3 (Applied Biosystems). The sequences of all primers used are listed in Supplementary Table 2. *GAPDH*, *HPRT*, and *ACTB* were used as internal controls for normalization.

**RNA-sequencing analysis.** Total RNA was extracted from A549, H460, and H2023 cells 48 h following transfection with control siRNA or NRF2 siRNA (cat. no. HSS107128 and HSS107130) using the RNeasy Mini Kit (Qiagen). Cells treated with the control siRNA were analyzed in biological duplicates. The cells treated with two different NRF2 siRNAs (HSS107128 and HSS107130) were analyzed separately and were considered to be biological duplicates of the NRF2-knockdown sample. Total RNA was extracted from ABC1 and HCC4006 cells 16 hrs after treatment with 100 µM DEM or vehicle (DMSO). Cells treated with either 100 µM DEM or vehicle (DMSO) were analyzed in biological duplicates. In all, 4 µg of total RNA from A549 cells was subjected to rRNA removal using the Ribo-Zero Gold kit (Illumina). cDNA sequencing libraries were then prepared using the SureSelect Strand-Specific RNA library preparation kit (Agilent Technologies) with a modified protocol omitting the polyA selection step. Total RNA from all other cell lines was used to prepare cDNA sequencing libraries using the SureSelect Strand-Specific RNA library preparation kit (Agilent Technologies) after the polyA selection step. The libraries were quantified by qMiSeq and sequenced on a NextSeq 500 (Illumina) to generate 80-base paired-end reads. Data analysis was performed on the Illumina BaseSpace platform (https://basespace.illumina.com) as follows. Raw fastq sequencing files were aligned to the hg19 reference genome using TopHat Alignment Version 1.0.0[59]. After read mapping, transcripts were assembled using Cufflinks software Version 2.3.1[59]. Expression level estimations were reported for each sample as fragments per kilobase of transcript sequence per million mapped fragments (FPKM). Raw and processed data were deposited in GEO (GSE118841, GSE118842). To identify NRF2 downstream effectors, significantly decreased genes by *NRF2* knockdown were selected for A549, H460, and H2023 cells, and significantly increased genes by DEM treatment were selected for ABC1 and HCC4006 cells. BH correction-adjusted *p*-values were calculated, and decreases and increases in gene expression were considered significant when *p*-values were less than 0.05.

**Analysis of gene expression in mouse tissues.** Wild-type C57BL/6 mice were treated with CDDO-Im or DMSO as a vehicle. CDDO-Im was administered at a dose of 30 µmol/kg, for which 3 mM (nmol/µl) working solution was prepared by diluting 30 mM (nmol/µl) CDDO-Im stock solution in DMSO with an isovolume of Cremophor-EL and 8x volume of PBS. The mice were sacrificed 6 h after the CDDO-Im administration, and representative organs were dissected. *Keap1*-knockdown (*Keap1* KD) and control wild-type mice that were obtained in the same breeding colony were sacrificed, and representative organs were dissected. The mice were used at between 7 and 9 weeks of age. Tissue samples were homogenized in ISOGEN (NIPPON GENE), and RNAs were purified. Reverse transcription and quantitative PCR were conducted in the same way as described for cell culture samples.

**Analysis of non-small cell lung cancer patients.** Meta-analysis of the TCGA database: RNA-seq data of 503 LUAD and 466 LUSC cases from TCGA, Pan-Cancer Atlas were analyzed. NRF2 scores were defined as average values of z-scores of 6 representative NRF2 target genes, *NQO1*, *SLC7A11*, *GCLC*, *GCLM*, *TXNRD1* and *NR0B1*. Mutation data were also obtained from 503 LUAD cases registered in TCGA, PanCancer Atlas.

Immunohistochemistry: Rabbit polyclonal antibodies against human NOTCH3 (ab60087, Abcam; 1:100), NRF2 (sc-13032X, Santa Cruz; 1:100) and CEBPB (SAB4500112, Sigma-Aldrich; 1:100) were used. A Histofine Kit (Nichirei Biosciences), which employs the streptavidin-biotin amplification method, was used in this study. Antigen retrieval was performed by heating slides in the microwave for 20 min (for NOTCH3) or by autoclaving slides for 5 min (for NRF2 and CEBPB) in citric acid buffer (2 mM citric acid and 9 mM trisodium citrate dehydrate (pH 6.0)). The antigen-antibody complex was visualized using 3,3'-diaminobenzidine (DAB) solution (1 mM DAB, 50 mM Tris–HCl buffer (pH 7.6), and 0.006% H2O2) and counterstained with hematoxylin. As negative controls, we used normal rabbit IgG instead of the primary antibody or no secondary antibody. No specific immunoreactivity was detected in these sections.

Immunoreactivity scoring: Immunohistological evaluation was blindly performed by a pathologist (T.S.). NRF2 immunoreactivity was detected mostly in the nucleus whereas NOTCH3 and CEBPB immunoreactivities were detected in the cytoplasm and/or nucleus. The immunoreactivities of these antibodies were categorized into three levels; ratios of positive carcinoma cells were <10%, between 10% and 50% and more than 50%. When tumor samples were classified into two groups, positive or negative, the threshold was set at 10%.

**Chromatin immunoprecipitation assay.** Chromatin immunoprecipitation (ChIP) was performed using anti-H3K27ac antibody (MABI0309, MAB Institute), anti-NRF2 antibody (#12721, Cell Signaling Technology), anti-FOSL2 antibody (#19967S, Cell Signaling Technology), anti-CEBPB antibody (sc-150 X, Santa Cruz) and rabbit IgG (#55944, Cappel/MP Biomedicals). Cells were cross-linked with 1% formaldehyde for 10 min and lysed. The samples were sonicated to shear the DNA using an ultrasonic disintegrator (VP-15S, Taitec) by 20 strokes of 50% duty cycle at output level 3–5 repeated for 8–12 times. The solubilized chromatin fraction was incubated with the primary antibodies overnight, which were prebound to anti-mouse IgG-conjugated Dynabeads (anti-H3K27ac antibody) or anti-rabbit IgG-conjugated Dynabeads (the rest of the antibodies) (Thermo Fisher Scientific). In all, 30 µl Dynabeads were used per sample with 2 µg (NRF2, FOSL2, and CEBPB) or 0.2 µg (H3K27ac) of antibodies. The same amount of rabbit IgG was used as a control sample. Precipitated DNA was de-crosslinked, purified and used for quantitative real-time PCR with the primers listed in Supplementary Table 3.

**ChIP-sequencing (ChIP-seq) analysis.** ChIP was performed using control or NRF2 siRNA-treated A549 cells with an anti-H3K27ac antibody (MABI0309, MAB Institute), as described above. Precipitated DNA was de-crosslinked, purified and used for library preparation. Sequencing libraries were prepared from 1.0 or 2.0 ng of ChIPed DNA and input samples using a Mondrian SP + system (Nugen) with an Ovation SP Ultralow DR Multiplex System (Nugen). The libraries were further purified and size-selected using an AMPure XP Kit (Beckman Coulter) and were quantified using a quantitative MiSeq (qMiSeq) method[60]. Optimally diluted libraries were sequenced on a HiSeq2500 (Illumina) to generate 101-base single-end reads. Sequencing files were aligned to the hg19 reference genome using Bowtie2 version 2.2.6[61]. Reads with a mapping quality <20 were removed using SAMTools version 1.3.1[62]. Peaks were called with MACS2 version 2.1.0.20151222[63]. Default parameters were used for these calculations. The ENCODE blacklis[64], obtained on March 22, 2016, was applied to all obtained peaks to filter out possible non-functional signals. The ChIP-seq results from three independent experiments were merged using WiggleTools version 1.2[65]. Each signal track was scaled by the number of total mapped reads, and these tracks were merged into a single data file based on their calculated mean values. Raw and processed data were deposited in GEO (GSE118840).

ChIP-seq data from the ENCODE project (ENCSR584GHV for NRF2, ENCSR541WQI for MAFK, ENCSR000BUB for CEBPB and the other transcription factors in A549 cells) were obtained as raw sequencing files and analyzed using the same procedure described above. Processed ChIP-seq data for H3K27ac in normal human lung samples were also obtained from the ENCODE database (ENCSR540ADS). ChIP-seq peak visualization was performed using the Integrative Genomic Viewer[66]. ENCODE ChIP-seq data for CEBPB, GR, MYC, MAX, and FOSL2 were accessed through the Integrative Genomic Viewer and visualized. To clarify binding occupancy for transcription factors and H3K27ac deposition, we used *_treat_pileup.bdg files generated by MACS2 with -- SPMR option. These profiles were converted into BigWig files by using KentUtils version 302 (available from https://github.com/ENCODE-DCC/kentUtils), then deepTools version 3.0.1[67] was adopted to draw aggregation plots and heat maps. We used BEDtools version v2.27.1[68] to identify overlapping between NRF2 and H3K27ac peaks.

**Correlation study of ENCODE ChIP-seq data using A549 cells.** To obtain binding profiles of transcription factors in A549 cells, we analyzed ENCODE ChIP-seq data by the method described in the previous section. The experiments with an inducible treatment like ethanol or dexamethasone were excluded since specific transcriptional responses can be observed. Jaccard index values and their significance for obtained peak call results against NRF2 peaks with H3K27ac marks were calculated by GenometriCorr package version 1.1.23[69].

**Production of virus particles.** For lentiviral infection, lentiviral and packaging vectors were transfected into 293FT cells. For retroviral infection, retroviral vectors were transfected into PLAT-A cells. The culture media were replaced with fresh media 24 h after transfection. The cells were incubated for an additional 24 h, and the culture supernatants were used as the lentivirus or retrovirus particle sources.

**Disruption of NOTCH3 enhancer to establish ΔN3U cells.** Two guide RNAs (gRNAs) were designed to disrupt NRF2 binding sites (antioxidant responsive elements; AREs) located 10-kb upstream of the transcription start site of the *NOTCH3* gene. Lentiviral vectors expressing these gRNAs together with Cas9 mRNA were constructed by inserting annealed oligoDNAs (Supplementary Table 4) into LentiCRISPRv2 (Addgene). H460, A549, and H2023 cells were infected with lentiviral particles with 12.5 µg/ml polybrene. After incubation for 24 hrs, the cells were re-plated in 10 cm dishes and incubated in selection medium containing 2 µg/ml puromycin. Single clones were selected using cloning rings (TOHO). DNA was purified from each clone, and the modified regions were amplified using the primer sets listed in Supplementary Table 5. The PCR products were cloned and sequenced to verify disruption of the NRF2 binding sites.

**Establishment of inducible NRF2 knockdown cells**. The inducible *NRF2* shRNA lentiviral vector (SMARTvector Inducible Lentiviral shRNA; V3SH11252, V3IHSMCG_6358637 and V3IHSMCG_6804335) and control vector (SMART-vector Inducible Non-targeting mCMV-TurboGFP; VSC11651) were purchased from Dharmacon. H460 cells were infected with lentiviral particles with 12.5 μg/ml polybrene. After 24 h, the cells were re-plated in 10 cm dishes and cultured in selection medium containing 2 μg/ml puromycin. In all, 10 μg/ml tetracycline (APOLLO) was added 48 h prior to cell harvest for immunoblot analysis. In all, 2 μg/ml doxycycline was added 48 h prior to initiation of the xenograft experiment using these cell lines.

**Introduction of LacZ and NOTCH3 intracellular domain (N3ICD) into ΔN3U cells**. Expression vectors for N3ICD (pLV[Exp]-CMV > {hNotch3ICD}:IRES:EGFP (ns):T2A:Bsd) and LacZ (pLV[Exp]-CMV > LacZ:IRES:EGFP(ns):T2A:Bsd) were purchased from Vector Builder. ΔN3U H460 cells were infected with lentiviral particles with 12.5 μg/ml polybrene. After incubation for 24 hrs, the cells were re-plated in 10 cm dishes and incubated in selection medium containing 2–5 μg/ml blasticidin.

**Introduction of wild-type KEAP1 into NRF2-activated NSCLC cells**. Adeno-viruses harboring expression vectors for human KEAP1 (ADV-212864) and GFP (SL100708) was purchased from Vector BIOLABS and SignaGen Laboratories, respectively. A549, H460 and H2023 cells were infected with 100 MOI of each adenovirus particle. After incubation for 72 hrs, the cells were harvested for each experiment.

**Establishment of KEAP1 Knockdown H23 Cells**. *Keap1* shRNA expression vector was generated by inserting a double-stranded DNA fragment, 5′-CCC **GCA AGG ACT ACC TGG TCA AGA** TTC AAG AGA **TCT TGA CCA GGT AGT CCT TGC** TTT TTA-3′ (a target sequence indicated in bold), into pSUPER-Retro vector (Oligoengine). For generation of *LacZ* shRNA expression vector as a control, a double-stranded DNA fragment, 5′-CCC **GCC CAT CTA CAC CAA CGT AAC** TTC AAG AGA **GTT ACG TTG GTG TAG ATG GGC** TTT TTA-3′ (a target sequence indicated in bold), was inserted into pSUPER-Retro vector (Oligoengine). H23 cells were infected with retroviral particles with 12.5 μg/ml polybrene. After 24 h, the cells were re-plated in 10 cm dishes and cultured in selection medium containing 2 μg/ml puromycin.

**Spheroid formation assay**. Cell growth was examined in spheroid culture. In all, $10^3$ cells with 100 μl culture media were seeded in each well of a low-attachment U-bottom plate (PrimeSurface 96 well Plate, MS-9096UZ, Sumitomo Bakelite Co.) followed by transfection with 2 pmol/well of siRNA using RNAiMAX (Invitrogen). siRNAs against *NRF2* (HSS107128), *F2RL2, FAM20B, GPI, GSTP1, HGD, HJURP, HMGA1, HPDL, NOTCH3, NRG4, PFN2, SRPK1, VRK1*, and control siRNA were used. Spheroids were observed 24 and 96 h after transfection. After 96 h, cell proliferation was assessed using the Cell Counting Kit-8 (Nacalai Tesque) according to the manufacturer's protocol.

**Oncosphere formation assay**. As one of the indicators of tumor-initiating activity, oncosphere formation was examined. Briefly, cells were cultured in ultra-low attachment dishes (Corning) in CSC medium. The CSC medium consisted of serum-free DMEM-F12 medium (Gibco-Invitrogen) containing 50 μg/ml insulin (Sigma-Aldrich), 0.4% Albumin Bovine Fraction V (Sigma-Aldrich), N-2 Plus Media Supplement (R&D Systems), Gibco B-27 Supplement (Thermo Fisher Scientific), 20 ng/ml EGF (Pepro Tech) and 10 ng/ml bFGF (Pepro Tech). For introduction of siRNAs against *NRF2* or *CEBPB*, A549 ($1 \times 10^4$ cells), H2023 ($4 \times 10^4$ cells for *NRF2*, $1 \times 10^4$ cells for *CEBPB*) and H460 ($1 \times 10^4$ cells) cells were transfected with siRNAs and directly seeded in CSC medium. The culture media were left unchanged until the cell harvest on day 7. For introduction of siRNA against *NOTCH3*, transfection of the siRNA was conducted in the regular culture condition. After 24 h, A549 ($2 \times 10^4$ cells), H2023 ($4 \times 10^4$ cells), and H460 ($2 \times 10^4$ cells) cells were reseeded in CSC medium. Culture media were left unchanged until the cell harvest on day 4 after reseeding. Viable cells were counted using trypan blue staining.

**Xenograft experiments**. H460, A549, and H2023 cell suspensions were combined with Matrigel (Corning) and subcutaneously injected into the flank of four-week-old male Balbc nu/nu mice. In the case of the inducible *NRF2*-knockdown H460 cells, $1 \times 10^4$ cells were injected and the resulting tumors were analyzed after 21 days. The recipient Balbc nu/nu mice were continuously treated with 1 mg/ml doxycycline in the drinking water containing 5% sucrose until they were sacrificed. In the case of the *NOTCH3* enhancer-disrupted (ΔN3E) H460 cells, $1 \times 10^3$ and $1 \times 10^4$ cells were injected and the resulting tumors were dissected and weighed after 21 days and 15 days, respectively. In the case of ΔN3E A549 and H2023 cells, $1 \times 10^5$ (A549) and $3 \times 10^5$ (H2023) cells were injected and both of the resulting tumors were dissected and weighed after 35 days. In the case of ΔN3E H460 cells with LacZ expression and those with N3ICD expression, $1 \times 10^4$ cells were injected and the resulting tumors were analyzed after 18 days. In the CDDP administration experiments, 0.3 mg/kg of CDDP or normal saline as a vehicle were injected into

the peritoneal cavity of the Balbc nu/nu mice once a week starting from the timing of H460 cell transplantation.

**Serial transplantation experiments**. Tumors from the primary xenograft experiment were dissected from mice and chopped into small pieces under sterile conditions and incubated at 37 °C for 1 h in 5 ml low glucose DMEM supplemented with 10% fetal bovine serum (FBS) and penicillin/streptomycin containing 1 mg/mL collagenase type IV (Sigma-Aldrich, #C5138) and 100 μg/ml DNase I (SIGMA). The samples were incubated with 5 ml RBC lysis buffer (155 mM $NH_4Cl$, 15 mM $NaHCO_3$, 0.1 M EDTA, pH7.3) for 10 min, followed by filtration to remove debris. The samples were then incubated with anti-biotinylated CD31 (#13-0311-82, eBioscience) and CD45 (#13-0451-85, eBioscience) antibodies, followed by reaction with Dynabeads M-280 streptavidin (Thermo Fisher Scientific) to remove murine cells of endothelial and hematopoietic origins. To further remove the remaining murine cells, the samples were reacted with anti-mouse MCH class I antibody (ab95572, Abcam), and human tumor cells were collected as the mouse MCH class I-negative fraction using flow cytometry (BD FACSAria, Becton Dickinson). Then, $1 \times 10^3$ WT and ΔN3E H460 cells were used for the secondary xenograft experiment. The time course of the experiment is shown in Supplementary Fig. 13d, e.

**Immunoprecipitation**. A549 cells were harvested using a cell scraper and washed three times in 1x PBS. The cell pellet was resuspended in 1x PBS containing 0.5 mM DTME and 0.5 mM DSP and incubated at room temperature for 30 min, followed by incubation in quenching buffer (20 mM Tris-HCl (pH 7.5), 5 mM cysteine) at 25 °C for 5 min. After washing in ice-cold PBS, the pellet was sonicated in RIPA buffer briefly and centrifuged at $14,000 \times g$ at 4 °C for 5 min. The supernatant was subjected to anti-NRF2 affinity purification. An anti-NRF2 antibody (#12721, Cell Signaling Technology) and control rabbit IgG (#55944, Thermo Fisher Scientific) were crosslinked to a 1:1 mixture of Dynabeads protein A and protein G (Thermo Fisher Scientific) with DMP and incubated with the supernatant at 4 °C for 2 h. After washing in RIPA buffer, the NRF2 complex was eluted from the beads by incubation in elution buffer (50 mM Tris-HCl (pH 8.0), 0.2 M NaCl, 2 w/v% SDS, 50 mM DTT) at 37 °C for 30 min. The eluate was analyzed by immunoblot analysis with anti-NRF2 antibody (sc-13032X, Santa Cruz; 1:1,000), anti-FOSL2 antibody (#19967S, Cell Signaling Technology; 1:1,000), and anti-CEBPB antibody (sc-150 X, Santa Cruz; 1:1,000).

**Quantification and statistical analysis**. Statistical significance was evaluated using an unpaired two-sample Student's *t*-test, Mann–Whitney test, the Wilcoxon rank sum test, and one-way ANOVA followed by the Bonferroni post hoc test. Confidence intervals were calculated for all fold change evaluations. Associations of NRF2/NOTCH3 statuses, NRF2/CEBPB statuses, and NRF2-CEBPB/NOTCH3 statuses were evaluated by a cross-table using a chi-square test. Kaplan-Meier analysis was performed for cumulative and relapse-free survival, and statistical significance was evaluated using the log-rank test. Univariate and multivariate analyses were evaluated using the Cox proportional hazards model. These analyses were performed using Microsoft Office Excel (Microsoft), Prism 7 (GraphPad Software, Inc.), JMP Pro 13 and StatView 5.0J software (SAS Institute). $P < 0.05$ and $\alpha < 0.05$ (confidence interval) were considered to be statistically significant.

**Reporting summary**. Further information on research design is available in the Nature Research Reporting Summary linked to this article.

## Data availability

RNA-seq data generated in this study have been deposited in GEO under the accession code GSE118841 and GSE118842. ChIP-seq data generated in this study have been deposited in GEO under the accession code GSE118840. RNA-seq data and mutation data of lung adenocarcinoma samples from TCGA, PanCancer Atlas are available at https://www.cbioportal.org/study/summary?id=luad_tcga_pan_can_atlas_2018. RNA-seq data of lung squamous cell carcinoma samples from TCGA, PanCancer Atlas are available at https://www.cbioportal.org/study/summary?id=lusc_tcga_pan_can_atlas_2018. ChIP-seq data of transcription factors in A549 cells from ENCODE are available at https://www.encodeproject.org/search/?searchTerm=A549&type=Experiment&assay_title=TF+ChIP-seq&limit=all. All the other data supporting the findings of this study are available within the article and its Supplementary Information files and from the corresponding authors upon reasonable request. Source data are provided with this paper.

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

## Acknowledgements

We thank Dr. Christian Siebel for critical reading of the manuscript, Dr. Hideyuki Saya for advice on cancer stemness and Drs. Kazuhiko Igarashi, Hiroki Shima and Kyoko Ochiai for advice on NRF2 complex purification. We also thank Drs. Shota Endo and Toshiyuki Takai for document preparation for clinical studies, Ms. Nozomi Hatanaka for deep sequencing technical support, Ms. Eriko Naganuma for histological sample preparaiton and the Biomedical Research Core of the Tohoku University Graduate School of Medicine for their technical support. This work was supported by JSPS under grant numbers 18H02621 (H.M), 18H04794 (H.M.), 17F17116 (H.M.), 17K08618 (H.S.), and 16K12519 (K.K.), the Naito Foundation (H.M.), a research grant from the Princess Takamatsu Cancer Research Fund 15-24728 (H.M.), the Uehara Memorial Foundation (H.M.), a research grant from the Gonryo Medical Foundation (H.S.) and AMED under grant numbers JP18am0101067 (K.K.) and JP20gm5010002 (H.M.). The funders had no role in the study design, data collection and analysis, decision to publish or paper preparation.

## Author contributions

K.O., H.A., Z.L., N.O., H.K., Y.Onodera, M.M.A., D.M., Takuma.S., F.K., and N.O. conducted the experiments, analyzed the data and wrote the paper. S.T. and I.M. analyzed the data. M.W., K.H., A.S., Y.Okada, and Takashi.S. analyzed the clinical data. M.Y. provided critical biomaterials and an analysis platform for the study and analyzed the data. K.K. supervised the research, analyzed data and wrote the paper. H.S. conducted the experiments, supervised the research, analyzed the data and wrote the paper. H.M. designed the study, supervised the research, analyzed the data and wrote the paper.

## Competing interests

The authors declare no competing interests.
