## [Peer Review File · Nature Communications]

REVIEWER COMMENTS

Reviewer #1 (Remarks to the Author):

NCOMMS-20-15879-T

In this paper, Okazaki and colleagues have addressed the question of whether hyperactivation of transcription factor NRF2 in tumour cells that harbour somatic mutations in KEAP1 or NFE2L2 (the gene encoding NRF2) contributes to the tumour-initiating and stem cell phenotype of aggressive cancer cells. The authors postulated that hyperactivation of NRF2 in non-small cell lung cancer (NSCLC) as a consequence of mutations in KEAP1 or NFE2L2 might result in the upregulation of a different battery of genes than those in which NRF2 is not constitutively activated, and that this might confer a tumour-initiating/stem cell phenotype. They first tested whether knockdown of NRF2 prevented A549, H460 and H2023 NSCLC cells, which harbour mutant KEAP1, from forming oncospheres when grown in culture under low attachment conditions. Then, using RNA-seq gene expression profiling of the A549, H460 and H2023 cells and also ABC1 and HCC4006 NSCLC cells with a normal KEAP1-NRF2 axis, the authors identified 36 genes that were upregulated in NSCLC with dysregulated NRF2 and could not be induced by DEM in cells with normal NRF2. Thereafter, the authors demonstrated that NOTCH3 is the key gene that hyperactivated NRF2 induces and is responsible for the tumour-initiating/stem cell phenotype, and that it does so through forming a complex with CEBPB. Most importantly, NOTCH3 is not inducible by pharmacological agents that activate NRF2 in normal mouse tissues or in tumour cells that possess NRF2 that is still subject to normal repression by KEAP1. Thus, this paper describes for the first time an NRF2-NOTCH3 axis that supports a tumour-initiating/stem cell phenotype and helps explain why somatic mutations in KEAP1 or NFE2L2 are associated with aggressive drug-resistant tumours.

The work described represents a genuine tour de force. In this reviewer's opinion, the paper is also extremely important because it allays fears that using inducing agents to activate NRF2 in order to provide cytoprotection [as in cancer chemoprevention] will somehow promote cancer. This paper contains a lot of new information that sheds new light on the involvement of NRF2 in tumourigenesis. It will be widely read, and well cited.

I have no major scientific concerns. However, given the large amount of data, there are inevitably points that need clarification. Also, a few typographical / grammatical points that might be changed.

Point 1. Lines 298, 370, 385: In several places throughout the text, mention is made that CEBPB "invigorates NRF2" and that cooperativity occurs between CEBPB and NRF2. It might be helpful to the reader if the nature of this invigoration/co-operativity was expanded upon or speculated. At present it seems somewhat nebulous. For example, at present, the text may give the reader the impression that CEBPB and NRF2 can form a heterodimer that allow upregulation of NOTCH3, and this conclusion would be supported by the cartoon depicted in Extended Figure 14. By contrast, Figure 7b shows in the flanking region of the NOTCH3 gene a CEBPB-binding site that is in close proximity to two or three ARE sequences, to which an NRF2-small MAF heterodimer would bind. Given these facts, it would appear that somehow or other CEBPB [presumably as a homo-dimer or hetero-dimer with another CEBP subunit], interacts with a NRF2-small MAF heterodimer to alter chromatin structure, and so provide access to the cognate CEBPB-binding site and the AREs. How CEBPB and NRF2 orchestrate these events is unclear, as the opening up of chromatin could involve other proteins. Nevertheless, it might help the reader if some temporal explanation of the sequence of events that the authors envisage were included in the Discussion section.

Point 2. There seems to be little comment about how NRF2 upregulates CEBPB. It has been reported by Hou et al (2012) Free Radic Biol Med 52: 462-472 that the CEBPB gene contains an ARE sequence. Is this correct? If so, maybe this is a paper worth citing.

Point 3. What other genes besides NOTCH3 are regulated by the invigoration/co-operative actions of CEBPB and NRF2? Is it known how many genes contain CEBPB-binding sites and ARE sequences in close proximity? For example, in line 300, mention is made of FAM20B, 2C3H12A and C5AR1. Do the regulatory regions of these genes contain CEBPB-binding and ARE sequence motifs?

Point 4. Lines 87 – 90: It might be helpful to the reader to define precisely what is meant by the term 'tumour-initiating cell' and how such cells arise. Providing precise terms and explanations that avoid jargon would be valuable because the term 'tumour-initiating cells' [which seems to imply recurrence after therapy] is distinct from the phrase 'initiation of carcinogenesis', at least as has been used in the older cancer literature. Moreover, the authors only cite Baylin (2011) in the context of 'tumour-initiating cell', and the article referred to is only a 2-page News & Views that gives little insight into what is a major theme of the current study.

Point 5. Lines 117 – 125: Knockdown of NRF2 in A549, H460 and H2023 cells inhibited stem-like features. What about ABC1 and HCC4006 cells? Did they display stem-like features and, if so, what happens when NRF2 is knocked down?

Point 6. Lines 145 – 148: If KEAP1 was knocked down in ABC1 and HCC4006 cells, are the same 123 genes upregulated in these two cell lines as were downregulated in A549, H460 and H2023 cells when NRF2 was knocked down? In view of its lack of specificity, why was DEM used to treat ABC1 and HCC4006 cells?

Point 7. Line 46: Suggest rewording as- '...among which the NOTCH3 enhancer was shown to be critical for promotion of...'

Point 8. Line 111: Suggest rewording as- 'pathological significance of the NRF2-NOTCH3 axis...'

Point 9. Line 121: Suggest rewording as '...to allow them to grow in the form...'

Point 10. Lines 124-125: Suggest rewording as '...suggesting that when activated, NRF2 promotes a stem-like phenotype in NSCLCs.'

Point 11. Line 184: Hyphenate 'publicly' and 'available', giving 'publicly-available'.

Point 12. Line 233: add 'an', giving '...whereas an apparent increase...'

Point 13. Line 260: add 'the', giving '...at the NROB1 locus...'

Point 14. Line 289: Check spelling of 'heterodimeric'.

Point 15. Line 299: Check spelling of 'dependant'.

Point 16. Lines 351-352: Suggest rewording as '...supporting the notion that the NOTCH3 enhancer mediates the effects of hyperactivated NRF2 on tumour growth.'

Point 17. Discussion section: Some of the paragraphs in this section merely reiterate the findings of the study and do not discuss them in the context of the findings of other researchers. Could save space, to discuss point 1 above.

Reviewer #2 (Remarks to the Author):

This is an interesting article, where the authors describe a mechanism by which lung cancer cell lines with a mutation in KEAP1 (major cause of NRF2 addiction) have activated the regulatory axis NRF2-NOTCH3. Fact which confers to the tumor cells important advantages, increasing proliferation and reprogramming properties. Accumulation of CEBPB is described as a prerequisite to establish the enhancer where the NOTCH3 plays a fundamental role. This is an extensive work combining in vitro studies, using an important set of technical approaches, in silico/public databases analysis and a small series of poorly characterized primary human lung tumors. This reviewer considers that in order to establish the real relevance of the mechanism described in a subset of lung cancer tumors, it is important to address important doubts.

Major points

1-Authors claim that they realized an "unbiased study to identify...." but after the RNA-seq study they decided directly to select NOTCH3 as the principal actor of this history, and deepen in their study, and not in any of the rest of 35 genes identified during the unbiased study. In addition, they claim that NOTCH3 is a unique downstream effector of NRF2 in NRF2-activated NSCLC. To establish their singularity and unique relevance in the studied process they should do some experiments with the rest of relevant genes in the three selected cell lines (i.e. shRNAi for specific identified selected genes, etc.), to justify that NOTCH3 choice are based on criteria of functional relevance and not that was just a directed selection.

At this point the reviewer considers that to improve the identification of the specific response during the RNA-seq study it will be important to incorporate in the analysis the response of the KEAP1 mutated lines also treated with DEM agent (canonical pathway), as they realized for the NRF2 non activated cells.

2-Following with this point, the study has been realized mostly focused on cell lines that have constitutively activated NRF2 by mutation in KEAP1. What happens regarding the mechanisms described throughout the article if KEAP1 wild-type is reconstituted in these cell lines (i.e. using an inducible system)? What happens with NOTCH3, NRF2 accumulation and CEBPB? (also comment in point 5) in this context.

3-One of the weak points of this work that should be improved in order to establish clinical relevance in lung cancer is to simultaneously determine the accumulation/expression levels of NRF2, NOTCH3 and CEBPB proteins in patient samples. In this sense, the expression levels probably should be blinded categorized in a more complex ranking that positivity based on expresses >10% (now used). Authors should better define the series of 41 LUADs used to confirm in silico TCGA results. So, it is important to know important data such as KEAP1 mutation status (NOTCH3/NRF2 overexpression cases, KEAP1 is mutated?), as other important driving mutations in lung cancer as (EGFR, K-ras, ALK, Met, etc..). These cases are naïve of treatment, or postchemotherapy, etc..... In sum, it is important to better define a series of tumors where validate the in vitro/in silico results.

4-NRF2-dependent enhancer is a common feature of NRF2-activated NSCLC cell lines (KEAP1 mutated) characterized by the NRF2 expression and enhancer acetylation. Enhancer abrogation by CRISPR in H460 cells reduce dramatically NOTCH3, being a specific NOTCH3 enhancer and did not affect the other two genes located in the region (EPHX3 and BRD4). But, how affect enhancer abrogation the cross reaction of the other NRF2-specific identified genes (determined by qPCR or Western blot)? In addition, it would also be very important to assess the role of the enhancer in response to chemotherapy treatments used in lung cancer. Effect that should be evaluated on the WT and paired CRISPR deleted (AN3U) cells.

5-To understand the mechanism, the authors realize chip analysis and they conclude that CEBPB

protein levels in NRF2-activated NSCLC cells were higher than those in NRF2-normal NSCLC lines, being CEBPB a critical factor to invigorate enhancer NRF2 in NRF2-activated NSCLC lines. In addition, by CEBPB knockdown, the authors show reduction of acetylation in three of 36 loci of the specific genes, suggesting that cooperativity among NRF2 and CEBPB explains a part of the unique enhancer formation in NRF2-activated NSCLs in vitro. To establish the relevance of these associations, authors should replicate part of these results of association in fresh primary tumor samples, (analyzing for example the levels of NRF2, CEBPB, acetylation, etc.). As mentioned before, it would be also important to investigate what happens when KEAP1 wild-type is reconstituted, (i.e., using an inducible system)

6-Respect to in vivo experiments, Have they histological differences (i.e, analyzed by H&E) in the different context that can explain the observed differences in the tumor growth?. Does the analysis of these xenografted tumors support the in vitro results?

Other points

7-Line 120: reference must to be add here or in the methodology section in which paper are described A549, H460 and H2023 mutations in KEAP1 and activation of NRF2. If there is no publication of this result, show as a supplement results KEAP1 mutations/NRF2 activation in these cell lines. Authors cite a paper from Justilien et al, which use the same methodology but they analyze other cell lines (H1299, H1703, ChagoK1, H520 and H1869).

8-Extended figure 1b. Due to variability in tumor volume/weight, it will be important to increase the time of tumor growth or the number (n) of animals to validate the results. On the other hand, they induce the cells in vitro and then inject them into the mice and did not treat animals with Dox. The Dox induction has a peak at 24h and lasts up to 72h, thus it is important to check how long NRF2 silencing lasts in cells injected into mice?. In xenografted tumors it will be check for inhibition of NRF2 expression at the end (sacrifice of the mice).

9-Figure 1a-c. Please improve the quality of images because differences are not well appreciated. Authors compare clones transfected with different vectors (one clone generated without a gene and two clones generated with a gene and induced). They should compare the differences in growth and viability for the same clone induced and uninduced with Dox. From the reviewer's point of view the transfection process itself may have caused differences in the behavior of the lines.

10-Line 136-138: Oncospheres formation is really an exclusive capacity only of the lines activated with NRF2, as the authors claim? This is an important point, because authors argue that the differently expressed genes between activated and normal cells are related to the fact that activated cells are tumor initiators and normal cells are not.

11-Line 163-164: Why are the cases organized by expression of the genes regulated by NRF2 and not by the expression of NRF2 itself, which is the criterion that has been used previously for functional studies in cell lines?

12-Lines 626-627: justify the use of tetracycline and not doxycycline (dox) and at a concentration 10 times higher than that recommended by the kit used.

13-Lines 659-661: In vivo experiments with H460 cells with inducible knockdown of NRF2. Were animals treated with Dox once the cells were injected? Until what period did the gene remain silent? At sacrifice, how was the expression of NRF2? Was the same clone of non-induced cells compared to induced cells? This would be the proper used control.

Reviewer #3 (Remarks to the Author):

In this work, Okazaki et al. studied the transcriptional network regulated by the NRF2 transcription factor which is frequently activated in non-small cell lung cancers (NSCLC). By combining ChIP-seq and transcriptomic analyses in cancer cell lines depleted of NRF2 or after transient induction of NRF2, the authors identified a NOTCH3 associated enhancer as a direct NRF2 target in aberrantly NRF2 expressing cancer cells. The clinical relevance of the NRF2-NOTCH3 regulatory axis was further demonstrated by a combination of a series of in vitro and in vivo studies. Overall the study is well conducted and the conclusions are supported by solid and relevant results.

Minor remarks

1. What are the molecular bases for the aberrant activation of NRF2 in a subset of NSCLC? Is it only due to KEAP1 genetic alterations? Are other direct or indirect mechanisms described? This should be clarified in the introduction.
2. How the NRF2 activity of the cell lines is defined in order to classify the cell lines into NRF2-activated or NRF2-normal?
3. Figure 2 assessed the expression of NOTCH3 in NRF2-activated LUADs, which were classified based on the expression of NRF2-target genes. It will be more accurate to directly compare the expression of NRF2 and NOTCH3 in LUAD patients.
4. Figure 3a: The sequences at the bottom of the panel should be defined in the legend.
5. Figure 6a: Explain that the rank was based on colocalization with NRF2.

Typo:

Page 12: heterodimetic = heterodimeric

Page 13: dependennt = dependent

Reviewer #1 (Remarks to the Author):

NCOMMS-20-15879-T

In this paper, Okazaki and colleagues have addressed the question of whether hyperactivation of transcription factor NRF2 in tumour cells that harbour somatic mutations in KEAP1 or NFE2L2 (the gene encoding NRF2) contributes to the tumour-initiating and stem cell phenotype of aggressive cancer cells. The authors postulated that hyperactivation of NRF2 in non-small cell lung cancer (NSCLC) as a consequence of mutations in KEAP1 or NFE2L2 might result in the upregulation of a different battery of genes than those in which NRF2 is not constitutively activated, and that this might confer a tumour-initiating/stem cell phenotype. They first tested whether knockdown of NRF2 prevented A549, H460 and H2023 NSCLC cells, which harbour mutant KEAP1, from forming oncospheres when grown in culture under low attachment conditions. Then, using RNA-seq gene expression profiling of the A549, H460 and H2023 cells and also ABC1 and HCC4006 NSCLC cells with a normal KEAP1-NRF2 axis, the authors identified 36 genes that were upregulated in NSCLC with dysregulated NRF2 and could not be induced by DEM in cells with normal NRF2. Thereafter, the authors demonstrated that NOTCH3 is the key gene that hyperactivated NRF2 induces and is responsible for the tumour-initiating/stem cell phenotype, and that it does so through forming a complex with CEBPB. Most importantly, NOTCH3 is not inducible by pharmacological agents that activate NRF2 in normal mouse tissues or in tumour cells that possess NRF2 that is still subject to normal repression by KEAP1. Thus, this paper describes for the first time an NRF2-NOTCH3 axis that supports a tumour-initiating/stem cell phenotype and helps explain why somatic mutations in KEAP1 or NFE2L2 are associated with aggressive drug-resistant tumours.

The work described represents a genuine tour de force. In this reviewer's opinion, the paper is also extremely important because it allays fears that using inducing agents to activate NRF2 in order to provide cytoprotection [as in cancer chemoprevention] will somehow promote cancer. This paper contains a lot of new information that sheds new light on the involvement of NRF2 in tumourigenesis. It will be widely read, and well cited.

It is a great honor to know that our study is highly evaluated by the reviewer. We thank the reviewer for encouraging comments.

I have no major scientific concerns. However, given the large amount of data, there are inevitably points that need clarification. Also, a few typographical / grammatical points that might be changed.

Point 1. Lines 298, 370, 385: In several places throughout the text, mention is made that CEBPB "invigorates NRF2" and that cooperativity occurs between CEBPB and NRF2. It might be helpful to the reader if the nature of this invigoration/co-operativity was expanded upon or speculated. At present it seems somewhat nebulous. For example, at present, the text may give the reader the impression that CEBPB and NRF2 can form a heterodimer that allow upregulation of NOTCH3, and this conclusion would be supported by the cartoon depicted in Extended Figure 14. By contrast, Figure 7b shows in the flanking region of the NOTCH3 gene a CEBPB-binding site that is in close proximity to two or three ARE sequences, to which an NRF2-small MAF heterodimer would bind. Given these facts, it would appear that somehow or other CEBPB [presumably as a homo-dimer or hetero-dimer with another CEBP subunit], interacts with a NRF2-small MAF heterodimer to alter chromatin structure, and so provide access to the cognate CEBPB-binding site and the AREs. How CEBPB and NRF2 orchestrate these events is unclear, as the opening up of chromatin could involve other proteins. Nevertheless, it might help the reader if some temporal explanation of the sequence of events that the authors envisage were included in the Discussion section.

This is a very important point. We thank the reviewer for this precious advice. We have included our discussion on mechanistic aspects of NRF2-CEBPB cooperation in Discussion (lines 400-416).

Point 2. There seems to be little comment about how NRF2 upregulates CEBPB. It has been reported by Hou et al (2012) Free Radic Biol Med 52: 462-472 that the CEBPB gene contains an ARE sequence. Is this correct? If so, maybe this is a paper worth citing.

In A549 cells, one of the NRF2-activated NSCLC cell lines, no apparent NRF2 binding was observed in the promoter region of *CEBPB* gene, which was different from the result described in the FRBM paper by Hou et al. Instead, we have identified an NRF2 binding site and an NRF2-dependent enhancer in approximately 25-kbp upstream of a transcription initiation site of *CEBPB* gene (Figure 1 only for reviewers). This enhancer contains a typical ARE “TGCTGAGACAC”. We surmise that this enhancer is responsible for NRF2-mediated upregulation of *CEBPB* gene in NSCLC cells. To test this possibility, we should delete this enhancer region by CRISPR-Cas9 system to see whether *CEBPB* expression is suppressed. However, we did not go that far in this study.

Figure 1. only for reviewers. ChIP-seq and RNA-seq profiles of A549 cells. NRF2 binding and NRF2-dependent H3K27ac deposition are found in approximately 25-kbp upstream of a transcription initiation site of *CEBPB* gene.

Point 3. What other genes besides NOTCH3 are regulated by the invigoration/co-operative actions of CEBPB and NRF2? Is it known how many genes contain CEBPB-binding sites and ARE sequences in close proximity? For example, in line 300, mention is made of FAM20B, 2C3H12A and C5AR1. Do the regulatory regions of these genes contain CEBPB-binding and ARE sequence motifs?

We have checked CEBPB binding and the presence of the consensus sequence for CEBPB in the 21 NRF2-activated NSCLC-specific NRF2-dependent enhancers. CEBPB binding was observed in all NRF2-activated NSCLC-specific NRF2-dependent enhancers except for four of those in *BEND6*, *GSTP1*, *HMGAI* and *NRG4* loci. CEBPB consensus sequences were found in the enhancers with CEBPB binding. NRF2 consensus sequences were found in all 21 NRF2-activated NSCLC-specific NRF2-dependent enhancers.

	CEBPB binding	CEBPB consensus	NRF2 consensus
BEND6	X	-	O
C1S	O	O	O
C5AR1	O	O	O
F2RL2	O	O	O
FAM20B	O	O	O
GPI	O	O	O
GSTP1	X	-	O
HECW1	O	O	O
HGD	O	O	O
HJURP	O	O	O
HMGA1	X	-	O
HPDL	O	O	O
LDLRAD3	O	O	O
NEDD4	O	O	O
NOTCH3	O	O	O
NR0B1	O	O	O
NRG4	X	-	O
PFN2	O	O	O
RP11-383H13.1 (MSC)	O	O	O
TMOD3	O	O	O
ZC3H12A	O	O	O

Search for CEBPB and NRF2 consensus sequence was conducted using JASPAR2020.

<http://jaspar.genereg.net> (JASPAR2020)

CEBPB : MA0466.1, MA0466.2

NFE2L2 : MA0150.1, MA0510.2

relative profile score, threshold 80%

Point 4. Lines 87 – 90: It might be helpful to the reader to define precisely what is meant by the term ‘tumour-initiating cell’ and how such cells arise. Providing precise terms and explanations that avoid jargon would be valuable because the term ‘tumour-initiating cells’ [which seems to imply recurrence after therapy] is distinct from the phrase ‘initiation of carcinogenesis’, at least as has been used in the older cancer literature. Moreover, the authors only cite Baylin (2011) in the context of ‘tumour-initiating cell’, and the article referred to is only a 2-page News & Views that gives little insight into what is a major theme of the current study.

We are sorry for insufficient explanation of “tumor-initiating cells (TICs)” and inappropriate citation. We have changed citations and added “cancer stem cells” for clarification of “tumor-initiating cells” (lines 92-95).

Point 5. Lines 117 – 125: Knockdown of NRF2 in A549, H460 and H2023 cells inhibited stem-like features. What about ABC1 and HCC4006 cells? Did they display stem-like features and, if so, what happens when NRF2 is knocked down?

NRF2-normal NSCLC cell lines can be also cultured in the stem cell medium and form oncospheres, in which tumor-initiating cells (TICs) expressing stem cell markers are enriched. When *NRF2* is knocked down, some of the NRF2-normal NSCLC cells exhibit reduced oncosphere formation, and others do not show apparent changes. These results imply that *NRF2* contributes to the TIC status in NRF2-normal NSCLC cells in the context-dependent manner.

Point 6. Lines 145 – 148: If KEAP1 was knocked down in ABC1 and HCC4006 cells, are the same 123 genes upregulated in these two cell lines as were downregulated in A549, H460 and H2023

cells when NRF2 was knocked down? In view of its lack of specificity, why was DEM used to treat ABC1 and HCC4006 cells?

We conducted DEM treatment for NRF2-normal NSCLC cell lines, ABC1 and HCC4006, to obtain a set of genes including NRF2 target genes that are induced by transient activation of NRF2. As for an effect of *KEAP1* inhibition in NRF2-normal NSCLC cells, we conducted *KEAP1* knockdown experiment using H23 cells, another NRF2-normal NSCLC cell line. Among the 123 genes, which were downregulated by *NRF2* knockdown in NRF2-activated NSCLC cell lines, canonical NRF2 targets were upregulated whereas NRF2-activated NSCLC-specific NRF2 targets were not. We have included the data as Extended Data Figure 7a.

Point 7. Line 46: Suggest rewording as- ‘...among which the NOTCH3 enhancer was shown to be critical for promotion of...’

Point 8. Line 111: Suggest rewording as- ‘pathological significance of the NRF2-NOTCH3 axis...’

Point 9. Line 121: Suggest rewording as ‘...to allow them to grow in the form...’

Point 10. Lines 124-125: Suggest rewording as ‘...suggesting that when activated, NRF2 promotes a stem-like phenotype in NSCLCs.’

Point 11. Line 184: Hyphenate ‘publicly’ and ‘available’, giving ‘publicly-available’.

Point 12. Line 233: add ‘an’, giving ‘...whereas an apparent increase...’

Point 13. Line 260: add ‘the’, giving ‘...at the NROB1 locus...’

Point 14. Line 289: Check spelling of ‘heterodimeric’.

Point 15. Line 299: Check spelling of ‘dependant’.

Point 16. Lines 351-352: Suggest rewording as ‘...supporting the notion that the NOTCH3 enhancer mediates the effects of hyperactivated NRF2 on tumour growth.’

We appreciate careful reading of the manuscript and detailed advice by the reviewer. We have corrected the wording and spelling as suggested by the reviewer.

Point 17. Discussion section: Some of the paragraphs in this section merely reiterate the findings of the study and do not discuss them in the context of the findings of other researchers. Could save space, to discuss point 1 above.

According to the suggestion by the reviewer, we have deleted the third paragraph of Discussion and reorganized the second paragraph by including our discussion on mechanistic aspects of NRF2-CEBPB cooperation in Discussion (lines 400-416).

Reviewer #2 (Remarks to the Author):

This is an interesting article, where the authors describe a mechanism by which lung cancer cell lines with a mutation in KEAP1 (major cause of NRF2 addiction) have activated the regulatory axis NRF2-NOTCH3. Fact which confers to the tumor cells important advantages, increasing proliferation and reprogramming properties. Accumulation of CEBPB is described as a prerequisite to establish the enhancer where the NOTCH3 plays a fundamental role. This is an extensive work combining in vitro studies, using an important set of technical approaches, in silico/public databases analysis and a small series of poorly characterized primary human lung tumors. This reviewer considers that in order to establish the real relevance of the mechanism described in a subset of lung cancer tumors, it is important to address important doubts.

We thank the reviewer for giving constructive comments.

Major points

1-Authors claim that they realized an “unbiased study to identify....” but after the RNA-seq study they decided directly to select NOTCH3 as the principal actor of this history, and deepen in their study, and not in any of the rest of 35 genes identified during the unbiased study. In addition, they claim that NOTCH3 is a unique downstream effector of NRF2 in NRF2-activated NSCLC. To establish their singularity and unique relevance in the studied process they should do some experiments with the rest of relevant genes in the three selected cell lines (i.e shRNAi for specific identified selected genes, etc.), to justify that NOTCH3 choice are based on criteria of functional relevance and not that was just a directed selection.

According to the advice from the reviewer, we have added a process for narrowing down the stemness regulator candidate to NOTCH3 (Extended Data Figure 2b,c). The 36 genes, NRF2-activated NSCLC-specific NRF2 downstream effectors, were examined for their correlation with NRF2 activity in clinical samples registered in TCGA database. Among them, 13 genes that exhibited good correlation with NRF2 activity were further examined for their contribution to cell growth monitored by spheroid growth in three different NRF2-activated NSCLC cell lines. Because we wanted to exclude factors that promote tumorigenesis by increasing cell proliferation ability rather than tumor-initiating ability, we selected genes whose knockdown did not impair the spheroid growth. NOTCH3 remained as a candidate during this process commonly in the three NRF2-activated NSCLC cell lines.

At this point the reviewer considers that to improve the identification of the specific response during the RNA-seq study it will be important to incorporate in the analysis the response of the KEAP1 mutated lines also treated with DEM agent (canonical pathway), as they realized for the NRF2 non activated cells.

We are sorry for our insufficient explanation. DEM is a KEAP1 inhibitor for activating NRF2 in NRF2-normal NSCLC cells. If you treat KEAP1-mutated cell lines with DEM, you will get gene expression profiles independent of NRF2. We have added a phrase for clarification (line 141).

2-Following with this point, the study has been realized mostly focused on cell lines that have constitutively activated NRF2 by mutation in KEAP1. What happens regarding the mechanisms described throughout the article if KEAP1 wild-type is reconstituted in these cell lines (i.e using an inducible system)? What happens with NOTCH3, NRF2 accumulation and CEBPB? (also comment in point 5) in this context.

Following the reviewer's comment, we have conducted wild-type KEAP1 reconstitution in NRF2-activated NSCLC cells possessing *KEAP1* mutations. KEAP1 introduction decreased NRF2 accumulation (new Extended Data Figure 4), *NOTCH3* mRNA expression (new Figure 4b) and CEBPB protein level (new Figure 6e).

3-One of the weak points of this work that should be improved in order to establish clinical relevance in lung cancer is to simultaneously determine the accumulation/expression levels of NRF2, NOTCH3 and CEBPB proteins in patient samples.

We thank the reviewer for this very important comment. We examined CEBPB accumulation in the 41 LUAD cases, which have been shown as new Figure 7c-f.

In this sense, the expression levels probably should be blinded categorized in a more complex ranking that positivity based on expresses >10% (now used). Authors should better define the series of 41 LUADs used to confirm in silico TCGA results.

As suggested by the reviewer, immunoreactivities of antibodies against NRF2, NOTCH3 and CEBPB were categorized into three levels; ratios of positive carcinoma cells were less than 10%, between 10% and 50% and more than 50% (new Figure 7d,e). When tumor samples were classified into two groups, positive or negative, the threshold was set at 10% (new Figure 2c-e and new Figure 7f). We could see statistically significant correlation among NRF2, NOTCH3 and CEBPB accumulation in the 41 LUAD cases.

So, it is important to know important data such as KEAP1 mutation status (NOTCH3/NRF2 overexpression cases, KEAP1 is mutated?), as other important driving mutations in lung cancer as (EGFR, K-ras, ALK, Met, etc..).

We thank the reviewer for this productive comment. We have analyzed relation among *NOTCH3* expression status, NRF2 activity and mutation status of representative cancer drivers (new Extended Data Figure 3a). LUAD samples in TCGA database were divided into 4 groups according to the *NOTCH3* expression status and NRF2 activity. Mutation ratios were mostly comparable among the four groups in the case of *ALK*, *MET* and *TP53*. *EGFR* mutation was enriched in the LUAD cases with low NRF2 activities and modestly enriched in those with low *NOTCH3* expression. *KRAS* mutation was enriched in LUAD with high *NOTCH3* expression irrespective of NRF2 activities, whereas *KEAP1* mutation was enriched in LUAD cases with high NRF2 activity and particularly with high *NOTCH3* expression.

These cases are naïve of treatment, or postchemotherapy, etc..... In sum, it is important to better define a series of tumors where validate the in vitro/in silico results.

The 41 lung adenocarcinoma (LUAD) cases examined in this study were randomly selected from the patient cohort we previously described (Inoue et al., *Cancer Sci* 103 (4) 760-766, 2012). They did not receive irradiation or chemotherapy before surgery. We have added the description in the materials and methods section, "Patients and tissue specimens".

4-NRF2-dependent enhancer is a common feature of NRF2-activated NSCLC cell lines (KEAP1 mutated) characterized by the NRF2 expression and enhancer acetylation. Enhancer abrogation by CRISPR in H460 cells reduce dramatically NOTCH3, being a specific NOTCH3 enhancer and did not affected the other two genes located in the region (EPHX3 and BRD4). But, how affect enhancer abrogation the cross reaction of the other NRF2-specific identified genes (determined by qPCR o Western blot)?

We thank the reviewer for the advice. As suggested by the reviewer, we examined representative NRF2-activated NSCLC-specific NRF2 downstream effector genes (new Extended Data Figure 5e). Disruption of *NOTCH3* enhancer did not make significant impacts on the expression of these genes.

In addition, it would also be very important to assess the role of the enhancer in response to chemotherapy treatments used in lung cancer. Effect that should be evaluated on the WT and paired CRISP deleted (AN3U) cells.

We conducted xenograft experiment using control H460 cells and *NOTCH3* enhancer-deficient H460 cells (Δ N3U H460 cells) treated with or without cisplatin (CDDP), which is shown in new Figure 9f (Figure 8f in the original manuscript). Anti-tumorigenic effect of CDDP treatment was augmented for Δ N3U H460 tumors, suggesting that *NOTCH3* enhancer inhibition has a synergistic effect with cytotoxic chemotherapy.

5-To understand the mechanism, the authors realize chip analysis and they conclude that CEBPB protein levels in NRF2-activated NSCLC cells were higher than those in NRF2-normal NSCLC lines, being CEBPB a critical factor to invigorate enhancer NRF2 in NRF2-activated NSCLC lines. In addition, by CEBPB knockdown, the authors show reduction of acetylation in three of 36 loci of the specific genes, suggesting that cooperativity among NRF2 and CEBPB explains a part of the unique enhancer formation in NRF2-activated NSCLs in vitro. To establish the relevance of these associations, authors should replicate part of these results of association in fresh primary tumor samples, (analyzing for example the levels of NRF2, CEBPB, acetylation, etc.). As mentioned before, it would be also important to investigate what happens when KEAP1 wild-type is reconstituted, (i.e., using an inducible system).

We thank the reviewer for the important advice. As suggested by the reviewer, we examined fresh primary tumor samples by immunoblot analysis. We could see that NRF2 accumulation is positively correlated with CEBPB accumulation (new Figure 7a,b).

We have also conducted wild-type KEAP1 reconstitution in NRF2-activated NSCLC cells possessing *KEAP1* mutations. KEAP1 introduction decreased NRF2 accumulation (new Extended Data Figure 4), CEBPB protein level (new Figure 6e) and H3K27ac deposition at *NOTCH3* enhancer (new Figure 3d).

6-Respect to in vivo experiments, Have they histological differences (i.e, analyzed by H&E) in the different context that can explain the observed differences in the tumor growth?. Does the analysis of these xenografted tumors support the in vitro results?

Histological characteristics of *NOTCH3* enhancer-deficient tumors are similar to those of xenograft tumors treated with γ -secretase inhibitor for inhibiting NOTCH proteins, which was previously reported (DOI: 10.1158/0008-5472.CAN-07-1022). One different point is the emergence of necrotic foci, which was described for the treatment with γ -secretase inhibitor in the previous study but not apparent for the *NOTCH3* enhancer disruption in our study (Figure 2 only for reviewers).

Figure 2. only for reviewers. HE staining of xenograft tumors of control H460 (left), Δ N3U H460 clone 1-2 (middle) and Δ N3U H460 clone 2-2 (right). The control H460 tumor show densely-packed appearance of cancer cells, whereas cell alignment is looser and more irregular in tumors of Δ N3U H460 cells.

Other points

7-Line 120: reference must to be add here or in the methodology section in which paper are described A549, H460 and H2023 mutations in KEAP1 and activation of NRF2. If there is no publication of this result, show as a supplement results KEAP1 mutations/NRF2 activation in these cell lines. Authors cite a paper from Justilien et al, which use the same methodology but they analyze other cell lines (H1299, H1703, ChagoK1, H520 and H1869).

We are sorry for inappropriate citation. We have cited a paper that describes *KEAP1* mutations in A549, H460 and H2023 cells (Saigusa et al., Cancer Sci 2020).

8-Extended figure 1b. Due to variability in tumor volume/weight, it will be important to increase the time of tumor growth or the number (n) of animals to validate the results.

As suggested by the reviewer, we increased the time for tumor growth, from 15 days to 21 days, and the number of animals, from 6 to 8, in the xenograft experiment shown as Extended Data Figure 1b.

On the other hand, they induce the cells in vitro and then inject them into the mice and did not treat animals with Dox. The Dox induction has a peak at 24h and lasts up to 72h, thus it is important to check how long NRF2 silencing lasts in cells injected into mice?

We are very sorry that methods for xenograft experiment with inducible *NRF2* knockdown cells were missing. Doxycycline treatment was continued until the recipient mice were sacrificed for analysis. We have added the description in Methods (lines 721-724).

In xenografted tumors it will be check for inhibition of NRF2 expression at the end (sacrifice of the mice).

We examined *NRF2* expression levels in control H460 cells and *NRF2*-knockdown H460 cells before and after transplantation. *NRF2* expression was reduced in *NRF2*-knockdown cells before the transplantation whereas the reduction almost disappeared in the xenograft tumors at the time of sacrifice of the recipient mice (Figure 3 only for reviewers). This was probably because *NRF2*-knockdown H460 cells with efficient reduction of *NRF2* expression had been outcompeted during tumorigenesis and those with less reduction of *NRF2* expression had been enriched.

Figure 3. only for reviewers. *NRF2* expression levels in H460 cells with and without *NRF2* knockdown (left) and in xenograft tumors of H460 cells with and without *NRF2* knockdown at the time of sacrifice (right).

9-Figure 1a-c. Please improve the quality of images because differences are not well appreciated. Authors compare clones transfected with different vectors (one clone generated without a gene and two clones generated with a gene and induced). They should compare the differences in growth and viability for the same clone induced and uninduced with Dox. From the reviewer's point of view the transfection process itself may have caused differences in the behavior of the lines.

We have changed photos for the oncosphere culture experiments according to the suggestion from the reviewer. These experiments were done using transient transfection of siRNA. “control siRNA” samples were transfected with scrambled siRNA, and “*NRF2* siRNA” samples were transfected with siRNA against *NRF2*. #28 and #30 indicate different siRNAs. The transfected cells were cultured in the stem cell media for oncosphere formation.

10-Line 136-138: *Oncospheres formation is really an exclusive capacity only of the lines activated with NRF2, as the authors claim? This is an important point, because authors argue that the differently expressed genes between activated and normal cells are related to the fact that activated cells are tumor initiators and normal cells are not.*

Oncosphere formation is not an exclusive capacity only of the *NRF2*-activated NSCLC cells. *NRF2*-normal NSCLCs can be cultured in the stem cell media for oncosphere formation as we described in the response to Point 5 comment from the reviewer 1.

We have not described that “the differently expressed genes between activated and normal cells are related to the fact that activated cells are tumor initiators and normal cells are not”. We are afraid that this comment sounds rather invalid.

11-Line 163-164: *Why are the cases organized by expression of the genes regulated by NRF2 and not by the expression of NRF2 itself, which is the criterion that has been used previously for functional studies in cell lines?*

We have improved Figure 2a by analyzing TCGA data set of both LUAD and LUSC for relation between *NRF2* activity and *NOTCH3* expression.

Because *NRF2* activity is mainly determined by its protein accumulation level rather than its mRNA level, we did not use *NRF2* mRNA levels for the analysis. Instead, we defined “*NRF2* score” based on the expression levels of typical *NRF2* target genes, *NQO1*, *SLC7A11*, *GCLC*, *GCLM*, *TXNRD1* and *NROB1*, as expression levels of *NRF2* target genes reflect *NRF2* activity as a transcription activator. We could see that high *NRF2* activity is correlated with high *NOTCH3* expression in LUAD and LUSC (new Figure 2a).

12-Lines 626-627: justify the use of tetracycline and not doxycycline (dox) and at a concentration 10 times higher than that recommended by the kit used.

We used tetracycline only for the induction of *NRF2* shRNA in the cell culture experiment. Concentration of the tetracycline in this experiment was indeed higher than the normally used concentration. Because drug metabolism and detoxification capacity are expected to be increased in *NRF2*-activated NSCLC cells such as H460 cell, we used higher concentration to ensure enough inhibition of *NRF2*. We did not see any apparent cytotoxicity at this concentration of tetracycline, 10 µg/ml, in H460 cell.

13-Lines 659-661: In vivo experiments with H460 cells with inducible knockdown of NRF2. Were animals treated with Dox once the cells were injected? Until what period did the gene remain silent? At sacrifice, how was the expression of NRF2?

These comments are similar to the comment 8.

Recipient mice were continuously treated with doxycycline until they were sacrificed. We examined *NRF2* expression at the time of sacrifice (see Figure 2. only for reviewers).

Was the same clone of non-induced cells compared to induced cells? This would be the proper used control.

We consider that a control sample for the xenograft experiment with doxycycline inducible system should be the one expressing irrelevant shRNA in response to doxycycline so that doxycycline is equally given to control and test samples.

Moreover, in this experiment, inducible *NRF2*-knockdown H460 cells were not cloned but used as a mass stable transformant cells to avoid influences of vector insertion to a specific locus. "*NRF2* shRNA2" and "*NRF2* shRNA3" indicate that these mass stable transformant cells express different shRNAs against *NRF2* in response to doxycycline.

Reviewer #3 (Remarks to the Author):

In this work, Okazaki et al. studied the transcriptional network regulated by the NRF2 transcription factor which is frequently activated in non-small cell lung cancers (NSCLC). By combining ChIP-seq and transcriptomic analyses in cancer cell lines depleted of NRF2 or after transient induction of NRF2, the authors identified a NOTCH3 associated enhancer as a direct NRF2 target in aberrantly NRF2 expressing cancer cells. The clinical relevance of the NRF2-NOTCH3 regulatory axis was further demonstrated by a combination of a series of in vitro and in vivo studies. Overall the study is well conducted and the conclusions are supported by solid and relevant results.

We are very happy to know that the reviewer appreciates our work. We thank the reviewer for encouraging comments.

Minor remarks

1. What are the molecular bases for the aberrant activation of NRF2 in a subset of NSCLC? Is it only due to KEAP1 genetic alterations? Are other direct or indirect mechanisms described? This should be clarified in the introduction.

Aberrant activation of NRF2 in cancer cells can be caused by several mechanisms, including somatic mutations in *KEAP1* or *NRF2* genes, exon skipping in *NRF2* gene, epigenetic suppression of *KEAP1* gene, sequestration of KEAP1 protein to autophagosome by p62/SQSTM1, electrophilic attack of KEAP1 thiols by fumarate. We have referred to these examples in Introduction (lines 73-77).

2. How the NRF2 activity of the cell lines is defined in order to classify the cell lines into NRF2-activated or NRF2-normal?

In this study, NRF2-activated NSCLC cell lines were defined as those with *KEAP1* mutations, and NRF2-normal NSCLC cell lines were defined as those with intact KEAP1-NRF2 system. But as pointed out by the reviewer in the comment 1, *KEAP1* mutations are not the only cause of aberrant activation of NRF2. So, when we evaluate the NRF2 activity of primary tumors in TCGA database, we defined “NRF2 score” as average expression of 6 representative NRF2 target genes, *NQO1*, *GCLC*, *GCLM*, *SLC7A11*, *TXNRD1* and *NR0B1*, which reflects the level of NRF2-dependent transcription.

3. Figure 2 assessed the expression of NOTCH3 in NRF2-activated LUADs, which were classified based on the expression of NRF2-target genes. It will be more accurate to directly compare the expression of NRF2 and NOTCH3 in LUAD patients.

Because NRF2 activity is mainly determined by its protein accumulation level rather than its mRNA level, we did not use NRF2 mRNA levels for the analysis. Instead, we defined “NRF2 score” based on the expression levels of typical NRF2 target genes, *NQO1*, *SLC7A11*, *GCLC*, *GCLM*, *TXNRD1* and *NR0B1*, as expression levels of NRF2 target genes reflect NRF2 activity as a transcription activator. In the revised manuscript, we have conducted more detailed analysis of the relation between NRF2 activity and NOTCH3 expression in lung adenocarcinoma (LUAD) and squamous cell carcinoma (LUSC). We could see that high NRF2 activity is correlated with high *NOTCH3* expression both in LUAD and LUSC (new Figure 2a).

4. Figure 3a: The sequences at the bottom of the panel should be defined in the legend.

We have added definition of the sequence and arrows in the legend to Figure 3a.

5. *Figure 6a: Explain that the rank was based on colocalization with NRF2.*

We have added explanation that higher rank indicates better colocalization with NRF2 in the legend to Figure 6a.

Typo:

Page 12: heterodimetic = heterodimeric

Page 13: dependennt = dependent

We have corrected the spellings. We appreciate careful reading by the reviewer.

REVIEWERS' COMMENTS

Reviewer #1 (Remarks to the Author):

Thank you for answering all the questions and points raised in my report in a thorough manner. I am fully satisfied.

Reviewer #2 (Remarks to the Author):

The authors have carried out an important number of experiments on the questions raised. They have convincingly answered most of the points suggested by the reviewer. After newly reviewing the article with the changes made to the text and figures, I have no major scientific concerns.

Reviewer #3 (Remarks to the Author):

The authors have satisfactorily addressed all my questions. I do not have further remarks

REVIEWERS' COMMENTS

Reviewer #1 (Remarks to the Author):

Thank you for answering all the questions and points raised in my report in a thorough manner. I am fully satisfied.

Reviewer #2 (Remarks to the Author):

The authors have carried out an important number of experiments on the questions raised. They have convincingly answered most of the points suggested by the reviewer. After newly reviewing the article with the changes made to the text and figures, I have no major scientific concerns.

Reviewer #3 (Remarks to the Author):

The authors have satisfactorily addressed all my questions. I do not have further remarks.